# Derivative-enhanced Deep Operator Network

**Yuan Qiu, Nolan Bridges, Peng Chen**
Georgia Institute of Technology, Atlanta, GA 30332
{yuan.qiu, bridges, pchen402}@gatech.edu

## Abstract

The deep operator networks (DeepONet), a class of neural operators that learn mappings between function spaces, have recently been developed as surrogate models for parametric partial differential equations (PDEs). In this work we propose a derivative-enhanced deep operator network (DE-DeepONet), which leverages derivative information to enhance the solution prediction accuracy and provides a more accurate approximation of solution-to-parameter derivatives, especially when training data are limited. DE-DeepONet explicitly incorporates linear dimension reduction of high dimensional parameter input into DeepONet to reduce training cost and adds derivative loss in the loss function to reduce the number of required parameter-solution pairs. We further demonstrate that the use of derivative loss can be extended to enhance other neural operators, such as the Fourier neural operator (FNO). Numerical experiments validate the effectiveness of our approach.

## 1 Introduction

Using neural networks to approximate the maps between functions spaces governed by parametric PDEs can be very beneficial in solving many-query problems, typically arising from Bayesian inference, optimization under uncertainty, and Bayesian optimal experimental design. Indeed, once pre-trained on a dataset, neural networks are extremely fast to evaluate given unseen inputs, compared to traditional numerical methods like the finite element method. Recently various neural operators are proposed to enhance the learning capacity, with two prominent examples deep operator network (DeepONet) [1] and Fourier neural operator (FNO) [2], which are shown to be inclusive of each other in their more general settings [3, 4], see also their variants and other related neural operator architectures in [5, 6, 7, 8, 9, 10]. Though these work demonstrate to be successful in approximating the output function, they do not necessarily provide accurate approximation of the derivative of the output with respect to the input, which are often needed for many downstream tasks such as PDE-constrained optimization problems for control, inference, and experimental design [11, 12, 13, 14].

In this paper, we propose to enhance the performance of DeepONet through derivative-based dimension reduction for the function input inspired by [15, 16, 17, 18] and the incorporation of derivative information in the training to learn both the output and its derivative with respect to the input inspired by [19, 20]. These two derivative-enhanced approaches can significantly improve DeepONet's approximation accuracy for the output function and its directional derivative with respect to the input function, especially when the training samples are limited. We provide details on the computation of derivative labels of the solution of PDEs in a general form as well as the derivative-based dimension reduction to largely reduce the computational cost. We demonstrate the effectiveness of our proposed method (DE-DeepONet) compared to three other neural operators, including DeepONet, FNO, and derivative-informed neural operator (DINO) [20], in terms of both test errors and computational cost. In addition, we apply derivative learning to train the FNO and also compare its performance with other methods. The code for data generation, model training and inference, as well as configurations to reproduce the results in this paper can be found at https://github.com/qy849/DE-DeepONet.

38th Conference on Neural Information Processing Systems (NeurIPS 2024).

## 2 Preliminaries

This section presents the problem setup, high-fidelity approximation using finite element for finite dimensional discretization, and the DeepONet architecture in learning the solution operator.

### 2.1 Problem setup

Let $\Omega \subset \mathbb{R}^d$ denote an open and bounded domain with boundary $\partial\Omega \subset \mathbb{R}^{d-1}$, where the dimension $d = 1, 2, 3$. We consider a PDE of the general form defined in $\Omega$ as

$$\mathcal{R}(m, u) = 0, \tag{1}$$

prescribed with proper boundary conditions. Here $m \in V^{\text{in}}$ is an input parameter function defined in a separable Banach space $V^{\text{in}}$ with probability measure $\nu$ and $u \in V^{\text{out}}$ is the output as the solution of the PDE defined in a separable Banach space $V^{\text{out}}$. Our goal is to construct a parametric model $\hat{u}(m; \theta)$ to approximate the solution operator that maps the parameter $m$ to the solution $u$.

Once constructed, the parametric model $\hat{u}(m; \theta)$ should be much more computationally efficient to evaluate compared to solving the PDE with high fidelity approximation.

### 2.2 High fidelity approximation

For the high fidelity approximation of the solution, we consider using a *finite element method* [21] in this work. We partition the domain $\Omega$ into a finite set of subregions, called cells or elements. Collectively, these cells form a *mesh* of the domain $\Omega$. Let $h$ represent the diameter of the largest cell. We denote $V_h^{\text{in}}$ indexed by $h$ as the finite element space for the approximation of the input space $V^{\text{in}}$ with Lagrange basis $\{\phi_1^{\text{in}}, \cdots, \phi_{N_h^{\text{in}}}^{\text{in}}\}$ of dimension $N_h^{\text{in}}$ such that $\phi_i^{\text{in}}(x^{(j)}) = \delta_{ij}$ at the finite element node $x^{(j)}$, with $\delta_{ij}$ being the Kronecker delta function. Similarly, we denote $V_h^{\text{out}}$ as the finite element space for the approximation of the solution space $V^{\text{out}}$ with basis $\{\phi_1^{\text{out}}, \cdots, \phi_{N_h^{\text{out}}}^{\text{out}}\}$ of dimension $N_h^{\text{out}}$. Note that for the approximation to be high fidelity, $N_h^{\text{in}}$ and $N_h^{\text{out}}$ are often very large. To this end, we can write the high fidelity approximation of the input and output functions as

$$m_h(x) = \sum_{i=1}^{N_h^{\text{in}}} m_i \phi_i^{\text{in}}(x) \text{ and } u_h(x) = \sum_{i=1}^{N_h^{\text{out}}} u_i \phi_i^{\text{out}}(x),$$

with coefficient vectors $\boldsymbol{m} = (m_1, \cdots, m_{N_h^{\text{in}}})^T \in \mathbb{R}^{N_h^{\text{in}}}$ and $\boldsymbol{u} = (u_1, \cdots, u_{N_h^{\text{out}}})^T \in \mathbb{R}^{N_h^{\text{out}}}$, whose entries are the *nodal values* of $m$ and $u$ at the corresponding nodes.

### 2.3 DeepONet

We briefly review the DeepONet architecture [1] with a focus on learning the solution operator of the PDE in Equation (1). To predict the evaluation of solution function $u$ at any point $x \in \Omega \cup \partial\Omega$ for any given input function $m$, [1] design a network architecture that comprises two separate neural networks: a trunk net $t(\cdot\,; \theta_t)$, which takes the coordinate values of the point $x$ at which we want to evaluate the solution function as inputs, and a branch net $b(\cdot\,; \theta_b)$, which takes the vector $\boldsymbol{m}$ encoding the parameter function $m$ as inputs. In [1], the vector $\boldsymbol{m}$ is the function evaluations at a finite set of fixed points $\{x^{(j)}\}_{j=1}^{N_h^{\text{in}}} \subseteq \Omega \cup \partial\Omega$, that is, $\boldsymbol{m} = (m(x^{(1)}), \cdots, m(x^{(N_h^{\text{in}})}))^T$, which corresponds to coefficient vector in the finite element approximation with Lagrange basis at the same nodes. If the solution function is scalar-valued, then both neural networks output vectors of the same dimensions. The prediction is obtained by taking the standard inner product between these two vectors and (optionally) adding a real-valued bias parameter, i.e,

$$\hat{u}(\boldsymbol{m}; \theta)(x) = \langle b(\boldsymbol{m}; \theta_b), t(x; \theta_t) \rangle + \theta_{\text{bias}},$$

with $\theta = (\theta_b, \theta_t, \theta_{\text{bias}})$. If the solution $u$ is vector-valued of $N_u$ components, i.e., $u = (u^{(1)}, \ldots, u^{(N_u)})$, as in our experiments, we can use one of the four approaches in [3] to construct the DeepONet. Specifically, for each solution component, we use the same outputs of branch net with dimension $N_b$ and different corresponding groups of outputs of trunk net to compute the inner product. More precisely, the solution $u^{(i)}$ of component $i = 1, \ldots, N_u$, is approximated by

$$\hat{u}^{(i)}(\boldsymbol{m}; \theta)(x) = \langle b(\boldsymbol{m}; \theta_b), t^{(i)}(x; \theta_t) \rangle + \theta_{\text{bias}}^{(i)}, \tag{2}$$

where $t^{(i)}(x; \theta_t) = t(x; \theta_t)[(i-1)N_b + 1 : iN_b]$, the vector slice of $t(x; \theta_t)$ with indices ranging from $(i-1)N_b + 1$ to $iN_b$, $i = 1, \ldots, N_u$.

For this construction, the outputs of the branch net can be interpreted as the coefficients of the basis learned through the trunk net. By partitioning the outputs of the trunk net into different groups, we essentially partition the basis functions used for predicting different components of the solution. The DeepONet is trained using dataset $\mathcal{D} = \{(m^{(i)}, u^{(i)})\}_{i=1}^{N_{\mathcal{D}}}$ with $N_{\mathcal{D}}$ samples, where $m^{(i)}$ are random samples independently drawn from $\nu$ and $u^{(i)}$ are the solution of the PDE (with slight abuse of notation from the vector-valued solution) at $m^{(i)}$, $i = 1, \ldots, N_{\mathcal{D}}$.

## 3 DE-DeepONet

The DeepONet uses the input parameter and output solution pairs as the labels for the model training. The approximation for the derivative of the solution with respect to the parameter (and the coordinate) are not necessarily accurate. However, in many downstream tasks such as Bayesian inference and experimental design, the derivatives are required. We consider incorporating the the Fréchet derivative $du(m; \cdot)$ for the supervised training of the DeepONet, which we call derivative-enhanced DeepONet (DE-DeepONet). By doing this we hope the optimization process can improve the neural network's ability to predict the derivative of the output function with respect to the input function. Let $\theta$ denote the trainable parameters in the DE-DeepONet. We propose to minimize the loss function

$$L(\theta) = \lambda_1 \mathbb{E}_{m \sim \nu} \|u(m) - \hat{u}(m)\|_{L^2(\Omega)}^2 + \lambda_2 \mathbb{E}_{m \sim \nu} \|du(m; \cdot) - d\hat{u}(m; \cdot)\|_{\text{HS}}^2, \tag{3}$$

where the $L^2(\Omega)$ norm of a square integrable function $f$ is defined as $\|f\|_{L^2(\Omega)}^2 = (\int_\Omega \|f(x)\|_2^2 \, dx)^{1/2}$, the Hilbert–Schmidt norm of an operator $T : H \to H$ that acts on a Hilbert space $H$ is defined as $\|T\|_{\text{HS}}^2 = \sum_{i \in I} \|Te_i\|_H^2$, where $\{e_i\}_{i \in I}$ is an orthonormal basis of $H$. Here, $\lambda_1, \lambda_2 > 0$ are hyperparameters that balance different loss terms.

The main challenge of minimizing the loss function (3) in practice is that with high fidelity approximation of the functions $m$ and $u$ using high dimensional vectors $\boldsymbol{m}$ and $\boldsymbol{u}$, the term $\|du(m; \cdot) - d\hat{u}(m; \cdot)\|_{\text{HS}}^2$ (approximately) becomes $\|\nabla_{\boldsymbol{m}} \boldsymbol{u} - \nabla_{\boldsymbol{m}} \hat{\boldsymbol{u}}\|_F^2$, where the Frobenius norm of a matrix $M \in \mathbb{R}^{m \times n}$ is defined as $\|M\|_F^2 = \sum_{i=1}^m \sum_{j=1}^n M_{ij}^2$. It is a critical challenge to both compute and store the Jacobian matrix at each sample as well as to load and use it for training since it is often very large with size $N_h^{\text{out}} \times N_h^{\text{in}}$, with $N_h^{\text{in}}, N_h^{\text{out}} \gg 1$.

To tackle this challenge, we employ dimension reduction for the high dimensional input vector $\boldsymbol{m}$. The reduced representation of $\boldsymbol{m}$ is given by projecting $\boldsymbol{m}$ into a low dimensional linear space spanned by a basis $\boldsymbol{\psi}_1^{\text{in}}, \ldots, \boldsymbol{\psi}_r^{\text{in}} \in \mathbb{R}^{N_h^{\text{in}}}$, with $r \ll N_h^{\text{in}}$, that is,

$$\widetilde{\boldsymbol{m}} = (\langle \boldsymbol{\psi}_1^{\text{in}}, \boldsymbol{m} \rangle, \ldots, \langle \boldsymbol{\psi}_r^{\text{in}}, \boldsymbol{m} \rangle)^T \in \mathbb{R}^r.$$

To better leverage the information of both the input probability measure $\nu$ and the map $u : m \to u(m)$, we consider the basis generated by active subspace method (ASM) using derivatives [22]. ASM identifies directions in the input space that significantly affect the output variance, or in which the output is most sensitive. See Section 3.3 for the detailed construction. In this case, the term $\mathbb{E}_{m \sim \nu} \|du(m; \cdot) - d\hat{u}(m; \cdot)\|_{\text{HS}}^2$ can be approximated by $\frac{1}{N_1 N_2} \sum_{i=1}^{N_1} \sum_{j=1}^{N_2} \|\nabla_{\widetilde{\boldsymbol{m}}} u(m^{(i)})(x^{(j)}) - \nabla_{\widetilde{\boldsymbol{m}}} \hat{u}(m^{(i)})(x^{(j)})\|_2^2$ with a small amount of functions $m^{(i)}$ sampled from input probability measure $\nu$ and points $x^{(j)}$ in the domain $\Omega$. Note that $\nabla_{\widetilde{\boldsymbol{m}}} \hat{u}(m^{(i)})(x^{(j)})$ is vector of size $r$, which is computationally feasible. For comparison, we also conduct experiments if the basis is the most commonly used Karhunen–Loève Expansion (KLE) basis.

We next formally introduces our DE-DeepONet, which uses dimension reduction for the input of the branch net and incorporates its output-input directional derivative labels as additional soft constraints into the loss function.

### 3.1 Model architecture

We incorporate dimension reduction into DeepONet to construct a parametric model for approximating the solution operator. Specifically, if the solution is scalar-valued (the vector-valued case can be constructed similar to (2)), the prediction is given by

$$\hat{u}(\boldsymbol{m}; \theta)(x) = \langle b(\widetilde{\boldsymbol{m}}; \theta_b), t(x; \theta_t) \rangle + \theta_{\text{bias}}, \tag{4}$$

where $\theta = (\theta_b, \theta_t, \theta_{\text{bias}})$ are the parameters to be learned. The branch net $b(\cdot; \theta_b)$ and trunk net $t(\cdot; \theta_t)$ can be chosen as an MLP, ResNet, etc. Note that the branch net takes a small vector of the projected parameter as input. We also apply the Fourier feature embeddings [5], defined as $\gamma(x) = [\cos(Bx), \sin(Bx)]$, to the trunk net, where each entry in $B \in \mathbb{R}^{m \times d}$ is sampled from a Gaussian distribution $\mathcal{N}(0, \sigma^2)$ and $m \in \mathbb{N}^+, \sigma \in \mathbb{R}^+$ are hyper-parameters.

## 3.2 Loss function

In practical training of the DE-DeepONet, we formulate the loss function as follows

$$
\begin{aligned}
L(\theta) = &\frac{\lambda_1}{N_{\mathcal{D}}} \sum_{i=1}^{N_{\mathcal{D}}} \text{err}(\{(\hat{u}(\boldsymbol{m}^{(i)}; \theta)(x^{(j)}), u(m^{(i)})(x^{(j)}))\}_{j=1}^{N_x}) \\
&+ \frac{\lambda_2}{N_{\mathcal{D}}} \sum_{i=1}^{N_{\mathcal{D}}} \text{err}(\{(\nabla_{\boldsymbol{m}} \hat{u}(\boldsymbol{m}^{(i)}; \theta)(x^{(j)})\Psi^{\text{in}}, \Phi^{\text{out}}(x^{(j)})(\nabla_{\boldsymbol{m}} \boldsymbol{u}(m^{(i)})\Psi^{\text{in}})\}_{j=1}^{N_x}),
\end{aligned}
\tag{5}
$$

where $\{x^{(j)}\}_{j=1}^{N_x}$ are the nodes of the mesh, $\Psi^{\text{in}} = (\boldsymbol{\psi}_1^{\text{in}}|\cdots|\boldsymbol{\psi}_r^{\text{in}})$ is the matrix collecting the nodal values of the reduced basis of the input function space, and $\Phi^{\text{out}}(x) = (\phi_1^{\text{out}}(x), \ldots, \phi_{N_h^{\text{out}}}^{\text{out}}(x))$ is the vector-valued function consisting of the finite element basis functions of the output function space. The $\text{err}(\{(\boldsymbol{a}^{(i)}, \boldsymbol{b}^{(i)})\}_{i=1}^n)$ denotes the relative error between any two groups of vectors $\boldsymbol{a}^{(i)}$ and $\boldsymbol{b}^{(i)}$, computed as

$$
\text{err}(\{(\boldsymbol{a}^{(i)}, \boldsymbol{b}^{(i)})\}_{i=1}^n) = \frac{(\sum_{i=1}^n \|\boldsymbol{a}^{(i)} - \boldsymbol{b}^{(i)}\|_2^2)^{1/2}}{\varepsilon + (\sum_{i=1}^n \|\boldsymbol{b}^{(i)}\|_2^2)^{1/2}},
$$

where $\varepsilon > 0$ is some small positive number to prevent the fraction dividing by zero.

In the following, we explain how to compute different loss terms in Equation (5)

- The first term is for matching the prediction of the parametric model $\hat{u}(\boldsymbol{m}^{(i)}; \theta)$ evaluated at any set of points $\{x^{(j)}\}_{j=1}^{N_x} \subseteq \Omega \cup \partial\Omega$ with the high fidelity solution $u(m^{(i)})$ evaluated at the same points. The prediction $\hat{u}(\boldsymbol{m}^{(i)}; \theta)(x^{(j)})$ is straightforward to compute using Equation (4). This involves passing the reduced branch inputs $\widetilde{m}^{(i)}$ and the coordinates of point $x^{(j)}$ into the branch net and trunk net, respectively. The label $u(m^{(i)})(x^{(j)})$ is obtained using finite element method solvers.

- The second term is for learning the directional derivative of the evaluation $u(x^{(j)})$ with respect to the input function $m$, in the direction of the reduced basis $\psi_1^{\text{in}}, \ldots, \psi_r^{\text{in}}$. It can be shown that

$$
\nabla_{\boldsymbol{m}} \hat{u}(\boldsymbol{m}^{(i)}; \theta)(x^{(j)})\Psi^{\text{in}} = \nabla_{\widetilde{\boldsymbol{m}}} \hat{u}(\boldsymbol{m}^{(i)}; \theta)(x^{(j)}).
$$

  Thus, the derivative of the outputs of the parametric model can be computed as the partial derivatives of the output with respect to the input of the branch net via automatic differentiation. On the other hand, the derivative labels

$$
\Phi^{\text{out}}(x^{(j)})(\nabla_{\boldsymbol{m}} \boldsymbol{u}(m^{(i)})\Psi^{\text{in}}) = (du(m^{(i)}; \psi_1^{\text{in}})(x^{(j)}), \ldots, du(m^{(i)}; \psi_r^{\text{in}})(x^{(j)}))
$$

  are obtained by first computing the Gateaux derivatives $du(m^{(i)}; \psi_1^{\text{in}}), \ldots, du(m^{(i)}; \psi_r^{\text{in}})$ and then evaluating them at $x^{(j)}$. See Appendix B.3 for details about the computation of the Gâteaux derivative $du(m; \psi)$.

- We initialize the loss weights $\lambda_1 = \lambda_2 = 1$ and choose a loss balancing algorithm called the self-adaptive learning rate annealing algorithm [23] to update them at a certain frequency. This ensures that the gradient of each loss term have similar magnitudes, thereby enabling the neural network to learn all these labels simultaneously.

**Remark.** *For the training, the computational cost of the second term of the loss function (and its gradients) largely depends on the number of points used in each iteration. To reduce the computational and especially memory cost, we can use a subset of points, $N_x^{batch} = \alpha N_x$, where $\alpha$ is small number between 0 and 1 (e.g., $\alpha = 0.1$ in our experiments), in a batch of functions, though their locations could vary among different batches in one epoch. We find that this approach has little impact on the prediction accuracy of the model when $N_x$ is large enough.*

### 3.3 Dimension reduction

Throughout the paper, we assume that the parameter functions $m$ are independently drawn from some Gaussian random field [24]. In particular, we consider the case where the covariance function is the Whittle-Matérn covariance function, that is, $m \overset{i.i.d.}{\sim} \mathcal{N}(\bar{m}, \mathcal{C})$, where $\bar{m}$ is the (deterministic) mean function and $\mathcal{C} = (\delta I - \gamma \Delta)^{-2}$ ($I$ is the identity and $\Delta$ is the Laplacian) is an operator such that the square root of its inverse, $\mathcal{C}^{-\frac{1}{2}}$, maps random function $(m - \bar{m})$ to Gaussian white noise with unit variance [25]. The parameters $\delta, \gamma \in \mathbb{R}^+$ jointly control the marginal variance and correlation length.

We consider two linear projection bases for dimension reduction of the parameter function.

**Karhunen–Loève Expansion (KLE) basis.** The KLE basis is optimal in the sense that the mean-square error resulting from a finite representation of the random field $m$ is minimized [26]. It consists of eigenfunctions determined by the covariance function of the random field. Specifically, an eigenfunction $\psi$ of the covariance operator $\mathcal{C} = (\delta I - \gamma \Delta)^{-2}$ satisfies the differential equation

$$\mathcal{C}\psi = \lambda\psi. \tag{6}$$

When solved using the finite element method, Equation (6) is equivalent to the following linear system (See Appendix A.2 for the derivation)

$$M^{\text{in}}(A^{\text{in}})^{-1}M^{\text{in}}(A^{\text{in}})^{-1}M^{\text{in}}\boldsymbol{\psi} = \lambda M^{\text{in}}\boldsymbol{\psi}, \tag{7}$$

where the $(i, j)$-entries of matices $A^{\text{in}}$ and $M^{\text{in}}$ are given by

$$A_{ij}^{\text{in}} = \delta\langle\phi_j^{\text{in}}, \phi_i^{\text{in}}\rangle + \gamma\langle\nabla\phi_j^{\text{in}}, \nabla\phi_i^{\text{in}}\rangle, \quad M_{ij}^{\text{in}} = \langle\phi_j^{\text{in}}, \phi_i^{\text{in}}\rangle.$$

Here, we recall that $V_h^{\text{in}}$ is the finite element function space of input function, $\phi_i^{\text{in}}, i = 1, \ldots, N_h^{\text{in}}$ for the finite element basis of $V_h^{\text{in}}$, and $\langle\cdot, \cdot\rangle$ for the $L^2(\Omega)$-inner product. We select the first $r$ (typically $r \ll N_h^{\text{in}}$) eigenfunctions $\psi_1^{\text{in}}, \ldots, \psi_r^{\text{in}}$ corresponding to the $r$ largest eigenvalues for the dimension reduction. Let $\Psi^{\text{in}} = (\boldsymbol{\psi}_1^{\text{in}}|\cdots|\boldsymbol{\psi}_r^{\text{in}})$ denote the corresponding nodal values of these eigenfunctions. Since the eigenvectors $\boldsymbol{\psi}_i^{\text{in}}$ are $M^{\text{in}}$-orthogonal (or equivalently, eigenfunctions $\psi_i^{\text{in}}$ are $L^2(\Omega)$-orthogonal), the reduced representation of $\boldsymbol{m}$ can be computed as $\widetilde{\boldsymbol{m}} = (\Psi^{\text{in}})^T M^{\text{in}}\boldsymbol{m}$, that is, the coefficients of the low rank approximation of $m$ in the linear subspace spanned by the eigenfunctions $\psi_1^{\text{in}}, \ldots, \psi_r^{\text{in}}$.

**Active Subspace Method (ASM) basis.** The active subspace method is a gradient-based dimension reduction method that looks for directions in the input space contributing most significantly to the output variability [27]. In contrast to the KLE basis, the ASM basis is more computationally expensive. However, since the ASM basis captures sensitivity information in the input-output map rather than solely the variability of the input space, it typically achieves higher accuracy in predicting the output than KLE basis. We consider the case where the output is a multidimensional vector [22], representing the nodal values of the output function $u$. The ASM basis $\psi_i$, $i = 1, \ldots, r$, are the eigenfunctions corresponding to the $r$ largest eigenvalues of the generalized eigenvalue problem

$$\mathcal{H}\psi = \lambda\mathcal{C}^{-1}\psi, \tag{8}$$

where the action of operator $\mathcal{H}$ on function $\psi$ is given by

$$\mathcal{H}\psi = \mathbb{E}_{m\sim\nu(m)}[d^*u(m; du(m; \psi))]. \tag{9}$$

Here, $du(m; \psi)$ is the Gâteaux derivative of $u$ at $m \in V_h^{\text{in}}$ in the direction of $\psi \in V_h^{\text{in}}$, defined as $\lim_{\varepsilon\to 0}(u(m + \varepsilon\psi) - u(m))/\varepsilon$, and $d^*u(m; \cdot)$ is the adjoint of the operator $du(m; \cdot)$. When solved using finite element method, Equation (8) is equivalent to the following linear system (See Appendix A.3 for the derivation)

$$H\boldsymbol{\psi} = \lambda C^{-1}\boldsymbol{\psi}, \tag{10}$$

where the action of matrix $H$ on vector $\boldsymbol{\psi}$ is given by

$$H\boldsymbol{\psi} = \mathbb{E}_{\boldsymbol{m}\sim\nu(\boldsymbol{m})}[(\nabla_{\boldsymbol{m}}\boldsymbol{u})^T M^{\text{out}}(\nabla_{\boldsymbol{m}}\boldsymbol{u})\boldsymbol{\psi}], \tag{11}$$

and the action of matrix $C^{-1}$ on vector $\boldsymbol{\psi}$ is given by

$$C^{-1}\boldsymbol{\psi} = A^{\text{in}}(M^{\text{in}})^{-1}A^{\text{in}}\boldsymbol{\psi}. \tag{12}$$

Here, $M^{\text{out}}$ denotes the mass matrix of the output function space, i.e., $M^{\text{out}}_{ij} = \langle \phi^{\text{out}}_j, \phi^{\text{out}}_i \rangle$. In practice, when solving Equation (10), we obtain its left hand side through computing

$$(\langle \mathcal{H}\psi, \phi^{\text{in}}_1 \rangle, \dots, \langle \mathcal{H}\psi, \phi^{\text{in}}_{N^{\text{in}}_h} \rangle)^T$$

and its right hand side through the matrix-vector multiplication in Equation (12) (See Appendix A.3 for details). Similar to the KLE case, let $\Psi^{\text{in}}$ denote the nodal values of $r$ dominant eigenfunctions. Since the eigenvectors $\boldsymbol{\psi}^{\text{in}}_i$ are $C^{-1}$-orthogonal, the reduced representation of $\boldsymbol{m}$ can be computed as $\widetilde{\boldsymbol{m}} = (\Psi^{\text{in}})^T C^{-1} \boldsymbol{m}$. We use a scalable *double pass randomized algorithm* [28] implemented in hIPPYlib to solve the generalized eigenproblems Equation (7) and Equation (10).

To this end, we present the computation of the derivative label and the action of $\mathcal{H}$ as follows.

**Theorem 1.** *Suppose the PDE in the general form of* (1) *is well-posed with a unique solution map from the input function $m \in V^{in}$ to the output function $u \in V^{out}$ with dual $(V^{out})'$. Suppose the PDE operator $\mathcal{R} : V^{in} \times V^{out} \to (V^{out})'$ is differentiable with derivatives $\partial_m \mathcal{R} : V^{in} \to (V^{out})'$ and $\partial_u \mathcal{R} : V^{out} \to (V^{out})'$, and in addition $\partial_u \mathcal{R}$ is invertible with invertible adjoint $(\partial_u \mathcal{R})^*$. Then the directional derivative $p = du(m; \psi)$ for any function $\psi \in V^{in}$, and an auxillary variable $q$ such that $d^*u(m; p) = -(\partial_m \mathcal{R})^* q$ can be obtained as the solution of the linearized PDEs*

$$(\partial_u \mathcal{R})p + (\partial_m \mathcal{R})\psi = 0, \tag{13}$$

$$(\partial_u \mathcal{R})^* q = p. \tag{14}$$

*Proof sketch.* We perturb $m$ with $\varepsilon\psi$ for any small $\varepsilon > 0$ and obtain $\mathcal{R}(m + \varepsilon\psi, u(m + \varepsilon\psi)) = 0$. Using Taylor expansion to expand it to the first order and letting $\varepsilon$ approach 0, we obtain Equation (13), where $p = du(m; \psi)$. Next we compute $d^*u(m; p)$. By Equation (13), we have $du(m; \psi) = -(\partial_u \mathcal{R})^{-1}(\partial_m \mathcal{R})\psi$. Thus, $d^*u(m; \psi) = -(\partial_m \mathcal{R})^*(\partial_u \mathcal{R})^{-*}p$. Then we can first solve for $q = (\partial_u \mathcal{R})^{-*}p$ and then compute $-(\partial_m \mathcal{R})^* q$. See Appendix A.1 for a full proof.

## 4 Experiments

In this section, we present experiment results to demonstrate the derivative-enhanced accuracy and cost of our method on two test problems of nonlinear vector-valued PDEs in comparison with DeepONet, FNO, and DINO. Details about the data generation and training can be found in Appendix B.

### 4.1 Input probability measure

In all test cases, we assume that the input functions $m^{(i)}$ are i.i.d. samples drawn from a Gaussian random field with mean function $\bar{m}$ and covariance operator $(\delta I - \gamma\Delta)^{-\alpha}$. We take $\alpha = 2$ in two space dimensions so the covariance operator is of trace class [29]. It is worth noting that the parameters $\gamma$ and $\delta$ jointly control the marginal variance and correlation length, for which we take values to have large variation of the input samples that lead to large variation of the output PDE solutions. In such cases, especially when the mapping from the input to output is highly nonlinear as in our test examples, a vanilla neural network approximations tend to result in relatively large errors, particularly with a limited number of training data. To generate samples from this Gaussian random field, we use a scalable (in mesh resolution) sampling algorithm implemented in hIPPYlib [28].

### 4.2 Governing equations

We consider two nonlinear PDE examples, including a nonlinear vector-valued hyperelasticity equation with one state (displacement), and a nonlinear vector-valued Navier–Stokes equations with multiple states (velocity and pressure). For the hyperelasticity equation, we consider an experimental scenario where a square of hyperelasticity material is secured along its left edge while a fixed upward-right force is applied to its right edge [30]. Our goal is to learn the map between (the logarithm of) the material's Young's modulus and its displacement. We also consider the Navier–Stokes equations that describes viscous, incompressible creeping flow. In particular, we consider the lid-driven cavity case where a square cavity consisting of three rigid walls with no-slip conditions and a lid moving with a tangential unit velocity. We consider that the uncertainty comes from the viscosity term and aim to predict the velocity field. See Appendix B.1 for details.

## 4.3 Evaluation metrics

We use the following three metrics to evaluate the performance of different methods. For the approximations of the solution $u(m)$, we compute the relative error in the $L^2(\Omega)$ norm and $H^1(\Omega)$ norm on the test dataset $\mathcal{D}_{\text{test}} = \{(m^{(i)}, u^{(i)})\}_{i=1}^{N_{\text{test}}}$, that is,

$$\frac{1}{N_{\text{test}}} \sum_{m^{(i)} \in \mathcal{D}_{\text{test}}} \frac{\|\hat{u}(m^{(i)}; \theta) - u(m^{(i)})\|_{L^2(\Omega)}}{\|u(m^{(i)})\|_{L^2(\Omega)}}, \quad \frac{1}{N_{\text{test}}} \sum_{m^{(i)} \in \mathcal{D}_{\text{test}}} \frac{\|\hat{u}(m^{(i)}; \theta) - u(m^{(i)})\|_{H^1(\Omega)}}{\|u(m^{(i)})\|_{H^1(\Omega)}},$$

where

$$\|u\|_{L^2(\Omega)} = \int_\Omega \|u(x)\|_2^2 \, \mathrm{d}x = \boldsymbol{u}^T M^{\text{out}} \boldsymbol{u}, \quad \|u\|_{H^1(\Omega)} = (\|u\|_{L^2(\Omega)}^2 + \|\nabla u\|_{L^2(\Omega)}^2)^{1/2}.$$

For the approximation of the Jacobian $du(m; \cdot)$, we compute the relative error (for the discrete Jacobian) in the Frobenius norm on $\mathcal{D}_{\text{test}}$ along random directions $\omega = \{\omega_i\}_{i=1}^{N_{\text{dir}}}$, that is,

$$\frac{1}{|\mathcal{D}_{\text{test}}|} \sum_{m^{(i)} \in \mathcal{D}_{\text{test}}} \frac{\|d\hat{u}(m^{(i)}; \omega) - du(m^{(i)}; \omega)\|_F}{\|du(m^{(i)}; \omega)\|_F},$$

where $\omega_i$ are samples drawn from the same distribution as $m$.

## 4.4 Main results

We compare the prediction errors measured in the above three evaluation metrics and computational cost in data generation and neural network training for four neural operator architectures, including DeepONet [1], FNO [2], DINO [20], and our DE-DeepONet. We also add experiments to demonstrate the performance of the FNO trained with the derivative loss (DE-FNO) and the DeepONet trained with input dimension reduction but without the derivative loss (DeepONet (ASM) or DeepONet (KLE)). For FNO, we use additional position embedding [4] that improves its approximation accuracy in our test cases. For DINO, we use ASM basis (v.s. KLE basis) as the input reduced basis and POD basis (v.s. output ASM basis [20]) as the output reduced basis, which gives the best approximation accuracy. For the input reduced basis of DE-DeepONet, we also test and present the results for both KLE basis and ASM basis.

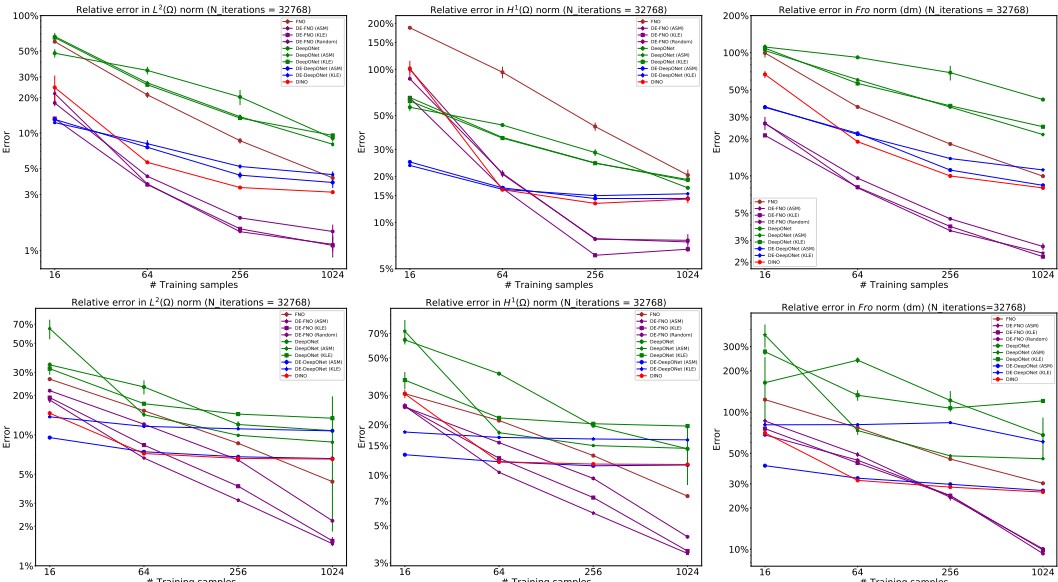

Figure 1: Mean relative errors ($\pm$ standard deviation) over 5 random seeds of neural network training for a varying number of training samples for the [top: hyperelasticity; bottom: Navier–Stokes] equation using different methods. Relative errors in the $L^2(\Omega)$ norm (left) and $H^1(\Omega)$ norm (middle) for the prediction of $u = (u_1, u_2)$. Right: Relative error in the Frobenius (Fro) norm for the prediction of $du(m; \omega) = (du_1(m; \omega), du_2(m; \omega))$.

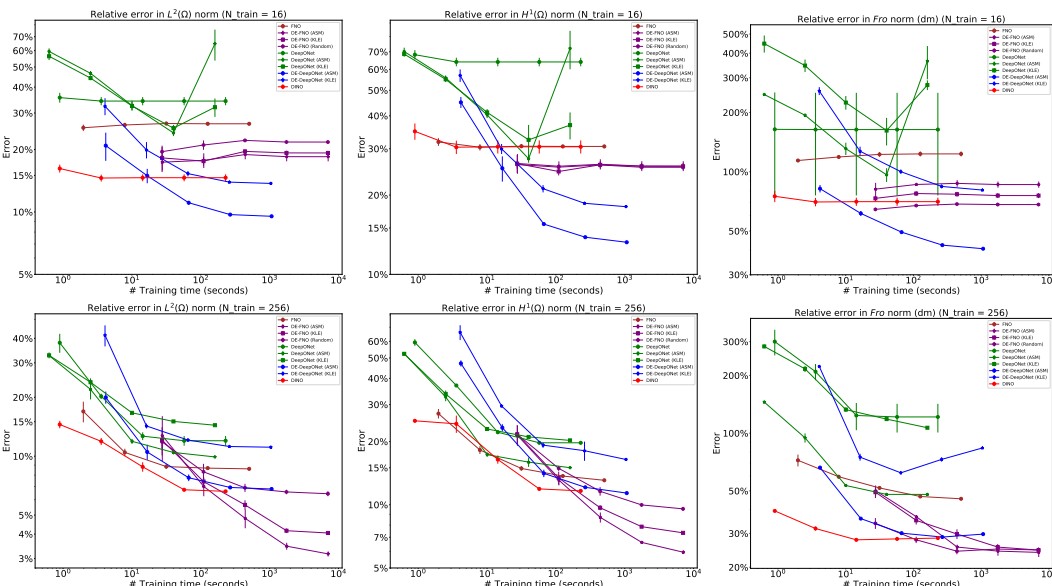

Figure 2: Mean relative errors ($\pm$ standard deviation) over 5 random seeds versus model training time for the Navier–Stokes equations when the number of training samples is [top: 16; bottom: 256].

**Test errors.** In Figure 1, we show a comparison of different methods for the hyperelasticity and Navier–Stokes problem in predicting the solution (in $L^2$ norm, left, and $H^1$ norm, middle) and its derivative (in Frobenius norm, right) with respect to parameter in the direction of $N_{\text{dir}} = 128$ test samples $\{\omega_i\}_{i=1}^{N_{\text{dir}}}$. First, we can observe significant improvement of the approximation accuracy of DE-DeepONet compared to the vanilla DeepONet for all three metrics in all cases of training samples. Second, we can see that FNO leads to larger approximation errors than DINO and DE-DeepONet for all metrics except when number of training samples is 1024. We believe that the reason why FNO performs better than DE-DeepONet and DINO when training samples are large enough is mainly due to the use of input dimensionality reduction in DE-DeepONet and DINO (where the linear reduction error cannot be eliminated by increasing training samples) whereas in FNO we use full inputs. We also see that DE-FNO performs the best among all models when the training samples are sufficient (256 or 1024), although in the compensation of much longer training time shown in Figure 2. Third, we can see that the approximation accuracy of DINO is similar to DE-DeepONet (ASM) but requires much longer inference time as shown in Table 5 (see Section 5 and Appendix B.3.2 for reasons). In DINO, the output reduced basis dimension is set to be smaller than or equal to the number of training samples as the output POD basis are computed from these samples, i.e., 16 for 16 samples and 64 for $\geq 64$ samples. Increasing the output dimension beyond 64 does not lead to smaller errors in our test. Finally, we can observe that DE-DeepONet using ASM basis leads to smaller errors than using KLE basis, especially for the Navier-Stokes problem.

Table 1: Output reconstruction error with 16 input reduced bases

|  | Relative $L^2$ error | |
| --- | --- | --- |
| Dataset | KLE | ASM |
| Hyperelasticity | 3.8% | 2.7% |
| Navier–Stokes | 17.4% | 5.8% |

Moreover, we present the output reconstruction error due to the input dimension reduction using KLE or ASM basis in Table 1. The errors provide the lower bound of the relative errors in $L^2(\Omega)$ norm of DINO and DE-DeepONet. We can see that using the ASM basis results in a lower output reconstruction error than the KLE basis (more significant difference observed in the more nonlinear Navier–Stokes equations). See Appendix B.6 for the decay of the reconstruction error with increasing number of reduced basis.

In addition, we provide full visualizations of the ground truth and prediction of both solution and derivatives in Appendix B.7. Visually, DE-DeepONet (ASM) consistently provides the best estimate (in terms of pattern and smoothness similarity with the ground truth) for both the solution and its derivative when the training data is limited (16 or 64 samples).

**Data generation computational cost.**  We use MPI and the finite element library FEniCS [31] to distribute the computational load of offline data generation to 64 processes for the PDE models considered in this work. See Table 2 for the wall clock time in generating the samples of Gaussian random fields (GRFs), solving the PDEs, computing the $r = 16$ reduced basis functions (KLE or ASM) corresponding to the 16 largest eigenvalues, generating the derivative labels, respectively. In Table 3, We also provide the total wall clock time of data generation of DE-DeepONet (ASM) (we only includes the major parts – computing high fidelity solution, ASM basis and dm labels [16 directions]) when $N_{\text{train}} = 16, 64, 256, 1024$ using 16 CPU processors.

Table 2: Wall clock time (in seconds) for data generation on $2 \times$ AMD EPYC 7543 32-Core Processors

| Dataset | Process | | | | |
|---|---|---|---|---|---|
| | GRFs $(N_{\text{all}} = 2000)$ (64 procs) | PDEs $(N_{\text{all}} = 2000)$ (64 procs) | KLE $(r = 16)$ (1 procs) | ASM $(r = 16)$ (16 procs) | dm labels $(N_{\text{all}} = 2000, r = 16)$ (64 procs) |
| Hyperelasticity | 1.1 | 9.7 | 0.4 | 1.4 | 19.5 |
| Navier–Stokes | 1.9 | 99.1 | 1.3 | 9.7 | 125.5 |

Table 3: Wall clock time (in seconds) for data generation with different number of training samples using 16 CPU processors

| Dataset\$N_{\text{train}}$ | PDEs + ASM basis + dm labels | | | |
|---|---|---|---|---|
| | 16 | 64 | 256 | 1024 |
| Hyperelasticity | 2 | 5 | 16 | 61 |
| Navier–Stokes | 17 | 38 | 124 | 470 |

Table 4: Wall clock time (seconds/iteration with batch size 8) for training on a single NVIDIA RTX A6000 GPU

| Dataset | Model | | | | |
|---|---|---|---|---|---|
| | DeepONet | FNO | DINO | DE-FNO | DE-DeepONet |
| Hyperelasticity | 0.007 | 0.015 | 0.007 | 0.215 | 0.022 |
| Navier–Stokes | 0.007 | 0.015 | 0.007 | 0.216 | 0.033 |

Table 5: Total wall clock time (in seconds) for each model inferring on 500 test samples of both the solution and dm in 128 random directions, using a single GPU and a single CPU (except where specified)

| Dataset | Model | | | | |
|---|---|---|---|---|---|
| | DeepONet | FNO/DE-FNO | DINO[1] 1 GPU + 1 CPU/16 CPUs | DE-DeepONet | Numerical solver 0 GPU + 16 CPUs |
| Hyperelasticity | 3 | 33 | 69/7 | 10 | 166 |
| Navier–Stokes | 3 | 33 | 2152/151 | 18 | 1103 |

[1] The inference time of DINO is dominated by the time required to compute evaluations of all finite element basis functions at the grid points using FEniCS (which may not be the most efficient, see Appendix B.3.2). Even though these grid points overlap with parts of the finite element nodes—allowing us to skip evaluations by extracting the relevant nodes—for a fairer comparison with DE-DeepONet (in terms of its ability to evaluate at any arbitrary point), we assume they are arbitrary points requiring explicit evaluation.

**Model training computational cost.**  We present comparisons of the wall clock time of each optimization iteration (with batch size 8) of different methods in Table 4 and convergence plot (error versus training time) in Figure 2 and the figures in Appendix B.5. We find that incorporating derivative loss leads to longer training time as expected. However, when the training data are limited, the increased computation cost is compensated for a significant reduction of errors. We note that there are potential ways to further reduce the training cost, e.g., by training the model with additional derivative loss only during the later stage of training, or by using fewer points for computing the derivative losses in each iteration. Additionally, thanks to the dimension reduction of the input, we can define a relatively small neural network and thus are able to efficiently compute the derivatives using automatic differentiation.

## 5  Related work

Our work is related to Sobolev training for neural networks [32], which was found to be effective in their application to model distillation/compression and meta-optimization. In the domain of surrogate

models for parametric partial differential equations, our work is more closely related to derivative-informed neural operator (DINO) [20] which is based on a derivative-informed projected neural network (DIPNet) [17], and presents an extension to enhance the performance of the DeepONet. Compared to DINO, although the DeepONet architecture (and its formulation of dm loss) requires longer training time, it offers the following advantages: (1) Potentially shorter inference time. The additional trunk net (which receives spatial coordinates) allows us to quickly query the sensitivity of output function at any point when input function is perturbed in any direction. While DINO can only provide the derivative of the output coefficients respect to the input coefficients (we call reduced dm), in order to compute the sensitivity at a batch of points, we need to post process the reduced dm by querying the finite element basis on these points and computing large matrix multiplications; (2) Greater flexibility and potential for improvements. Although both DeepONet and DINO approximate solution by a linear combination of a small set of functions, these functions together in DeepONet is essentially the trunk net, which is "optimized" via model training, whereas in DINO, they are POD or derivative-informed basis precomputed on training samples. When using DINO, if we encounter a case where the training samples not enough to accurately compute the output basis, the large approximation error between the linear subspace and solution manifold will greatly restrict the model prediction accuracy (see Figure 2 when $N_{\text{train}} = 16$). And the reduced dm labels only supports linear reduction of output. However, it is possible that we can further improve DeepONet by, e.g., adding physical losses (to enhance generalization performance) and Fourier feature embeddings (to learn high-frequency components more effectively) on the trunk net [5] and replacing the inner product of the outputs of two networks by more flexible operations [33, 9] (to enhance expressive power). The dm loss formulation of our work is broadly suitable any network architecture that has multiple subnetworks, where at least one of them receives high-dimensional inputs.

## 6  Discussion

In this work, we proposed a new neural operator–Derivative-enhanced Deep Operator Network (DE-DeepONet) to address the limited accuracy of DeepONet in both function and derivative approximations. Specifically, DE-DeepONet employs a derivative-informed reduced representation of input function and incorporates additional loss into the loss function for the supervised learning of the derivative of the output with respect to the inputs of the branch net. Our experiments for nonlinear PDE problems with high variations in both input and output functions demonstrate that adding this loss term to the loss function greatly enhances the accuracy of both function and derivative approximations, especially when the training data are limited. We also demonstrate that the use of derivative loss can be extended to enhance other neural operators, such as the Fourier neural operator.

We presented matrix-free computation of the derivative label and the derivative-informed dimension reduction for a general form of PDE problems by using randomized algorithms and linearized PDE solves. Thanks to this scalable approach, the computational cost in generating the derivative label data is shown to be only marginally higher than generating the input-output function pairs for the test problems, especially for the more complex Navier–Stokes equations which require more iterations in the nonlinear solves than the hyperelasticity equation.

**Limitations:**   We require the derivative information in the training and dimension reduction using ASM, which may not be available if the explicit form of the PDE is unknown or if the simulation only provides input-output pairs from some legacy code. Another limitation is that dimension reduction of the input function plays a key role in scalable data generation and training, which may not be feasible or accurate for intrinsically very high-dimensional problems such as high frequency wave equations. Such problems are also very challenging and remain unsolved by other methods to our knowledge.

## Acknowledgements

We would like to thank Jinwoo Go, Dingcheng Luo and Lianghao Cao for insightful discussions and helpful feedback.

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

# A Proofs

## A.1 Proof of Theorem 1

We first show how to compute $p$ given the PDE residual $\mathcal{R}(m, u) = 0$. Since $u$ is uniquely determined by $m$, we can write $u = u(m)$ so that $\mathcal{R}(m, u(m)) \equiv 0$ holds for any $m \in V^{\text{in}}$. Thus, for any $\varepsilon > 0$ and $\psi \in V^{\text{in}}$, we have

$$\mathcal{R}(m + \varepsilon\psi, u(m + \varepsilon\psi)) = 0.$$

Using the Taylor expansion we obtain

$$(\partial_m \mathcal{R}(m, u(m)))\varepsilon\psi + (\partial_u \mathcal{R}(m, u(m)))\delta u \approx 0, \tag{15}$$

where $\delta u = u(m + \varepsilon\psi) - u(m)$. Dividing both sides of Equation (15) by $\varepsilon$ and letting $\varepsilon$ approach $0$ yields

$$(\partial_m \mathcal{R}(m, u(m)))\psi + (\partial_u \mathcal{R}(m, u(m)))du(m; \psi) = 0.$$

For ease of notation, we write

$$\partial_m \mathcal{R} = \partial_m \mathcal{R}(m, u(m)), \quad \partial_u \mathcal{R} = \partial_u \mathcal{R}(m, u(m)).$$

Then $p$ is the solution to the linear PDE

$$(\partial_m \mathcal{R})\psi + (\partial_u \mathcal{R})p = 0. \tag{16}$$

We solve Equation (16) via its weak form

$$\langle (\partial_m \mathcal{R})\psi, v \rangle + \langle (\partial_u \mathcal{R})p, v \rangle = 0, \tag{17}$$

where $v$ is a test function in $V^{\text{out}}$.

Next we show how to compute $d^* u(m; p)$. By Equation (16), we have $du(m; \psi) = -(\partial_u \mathcal{R})^{-1}(\partial_m \mathcal{R})\psi$. Thus,

$$\begin{aligned} w := d^* u(m; p) &= (-(\partial_u \mathcal{R})^{-1}(\partial_m \mathcal{R}))^* p \\ &= -(\partial_m \mathcal{R})^*(\partial_u \mathcal{R})^{-*} p. \end{aligned} \tag{18}$$

Let $q := (\partial_u \mathcal{R})^{-*} p$. We can solve for $q$ via the weak form

$$\langle (\partial_u \mathcal{R})^* q, v \rangle = \langle p, v \rangle,$$

or equivalently,

$$\langle q, (\partial_u \mathcal{R})v \rangle = \langle p, v \rangle, \tag{19}$$

where $v$ is a test function in $V^{\text{out}}$.

By Equation (18), we have $w = -(\partial_m \mathcal{R})^* q$. For any test function $v \in V^{\text{in}}$, it holds that

$$\begin{aligned} \langle w, v \rangle &= \langle -(\partial_m \mathcal{R})^* q, v \rangle \\ &= \langle q, -(\partial_m \mathcal{R})v \rangle. \end{aligned} \tag{20}$$

Note that we do not need to solve for $w$ explicitly; we only compute $\langle w, v \rangle$ with $v$ as the finite element basis functions $\phi_1^{\text{in}}, \ldots, \phi_{N_h^{\text{in}}}^{\text{in}}$. The cost of computing the right hand side of Equation (20) arises from evaluating the directional derivative and its inner product with the finite element basis functions.

**Remark.** *By Equation (9), we use $N_{grad}$ samples $\{(m^{(i)}, u^{(i)})\}_{i=1}^{N_{grad}}$ to compute the Monte Carlo estimate of the action of operator $\mathcal{H}$ on any function $\psi \in V^{in}$, that is,*

$$\mathcal{H}\psi \approx \frac{1}{N_{grad}} \sum_{i=1}^{N_{grad}} d^* u(m^{(i)}; du(m^{(i)}; \psi)). \tag{21}$$

**Remark.** *When using the double pass randomized algorithm to obtain the first $r$ eigenpairs in Equation (10), we need to compute the action of $\mathcal{H}$ on $2(r + s)$ random functions in $V^{in}$ (e.g., their nodal values are sampled from the standard Gaussian distribution), where $s \in \mathbb{N}^+$ is typically a small oversampling parameter, often chosen between $5$ and $20$. To speed up the computation, we first compute the LU factorization of the matrices resulting from the discretization of the linear PDEs in Equation (17) and Equation (19). Then the action of $d^* u(m^{(i)}; du(m^{(i)}; \cdot))$ on these random functions can be efficiently computed via the forward and backward substitution. Furthermore, the computational time can be significantly reduced by parallelizing the computation of the average value in Equation (21) across multiple processors.*

## A.2 Proof of the equivalence of Equation (6) and Equation (7)

By the definition of $\mathcal{C} = (\delta I - \gamma \Delta)^{-2}$, Equation (6) is equivalent to

$$\psi = \lambda(\delta I - \gamma \Delta)^2 \psi. \tag{22}$$

We first compute $(\delta I - \gamma \Delta)\psi$. To do this, let $p = (\delta I - \gamma \Delta)\psi$ and multiply both sides of this equation by a test function $v$ and integrate

$$\int_\Omega p(x)v(x)\,\mathrm{d}x = \int_\Omega (\delta I - \gamma \Delta)\psi(x)v(x)\,\mathrm{d}x$$
$$= \delta \int_\Omega \psi(x)v(x)\,\mathrm{d}x + \gamma \int_\Omega \langle \nabla \psi(x), \nabla v(x) \rangle\,\mathrm{d}x, \tag{23}$$

where, in the second equality, we use integration by parts and the assumption that the test function vanishes on the boundary. Then we substitute $v$ with all of the finite element basis functions $\phi_1^{\mathrm{in}}, \dots, \phi_{N_h^{\mathrm{in}}}^{\mathrm{in}}$ and collect the corresponding linear equations for the nodal values of $p$

$$A^{\mathrm{in}}\boldsymbol{\psi} = M^{\mathrm{in}}\boldsymbol{p}, \tag{24}$$

where the $(i,j)$-entries of $A^{\mathrm{in}}$ and $M^{\mathrm{in}}$ are given by

$$A_{ij}^{\mathrm{in}} = \delta\langle \phi_j^{\mathrm{in}}, \phi_i^{\mathrm{in}}\rangle + \gamma\langle \nabla\phi_j^{\mathrm{in}}, \nabla\phi_i^{\mathrm{in}}\rangle, \quad M_{ij}^{\mathrm{in}} = \langle \phi_j^{\mathrm{in}}, \phi_i^{\mathrm{in}}\rangle.$$

By Equation (22), function $p$ satisfies $\psi = \lambda(\delta I - \gamma \Delta)p$. In the same manner we can see that

$$\lambda A^{\mathrm{in}}\boldsymbol{p} = M^{\mathrm{in}}\boldsymbol{\psi}. \tag{25}$$

Note that both matrices $M^{\mathrm{in}}$ and $A^{\mathrm{in}}$ are symmetric positive definite and thus nonsingular. Combining Equation (24) with Equation (25) yields

$$\lambda A^{\mathrm{in}}(M^{\mathrm{in}})^{-1}A^{\mathrm{in}}\boldsymbol{\psi} = M^{\mathrm{in}}\boldsymbol{\psi}, \tag{26}$$

or equivalently,

$$M^{\mathrm{in}}(A^{\mathrm{in}})^{-1}M^{\mathrm{in}}(A^{\mathrm{in}})^{-1}M^{\mathrm{in}}\boldsymbol{\psi} = \lambda M^{\mathrm{in}}\boldsymbol{\psi}. \tag{27}$$

## A.3 Proof of the equivalence of Equation (8) and Equation (10)

By Equation (8), for any test function $v \in V_h^{\mathrm{in}}$, it holds that

$$\langle \mathcal{H}\psi, v\rangle = \langle \lambda \mathcal{C}^{-1}\psi, v\rangle. \tag{28}$$

In particular, for any $m \sim \nu(m)$, as we let $v$ go through all of the finite element basis functions $\phi_i^{\mathrm{in}} \in V_h^{\mathrm{in}}$, we can show that

$$\begin{pmatrix} \langle d^*u(m; du(m;\psi)), \phi_1^{\mathrm{in}}\rangle \\ \vdots \\ \langle d^*u(m; du(m;\psi)), \phi_{N_h^{\mathrm{in}}}^{\mathrm{in}}\rangle \end{pmatrix} = (\nabla_{\boldsymbol{m}}\boldsymbol{u})^T M^{\mathrm{out}}(\nabla_{\boldsymbol{m}}\boldsymbol{u})\boldsymbol{\psi}. \tag{29}$$

Indeed, by the definition of Gateaux derivative, we have

$$du(m;\psi) = \lim_{\varepsilon \to 0} \frac{u(m + \varepsilon\psi) - u(m)}{\varepsilon}$$
$$= \lim_{\varepsilon \to 0} \sum_{i=1}^{N_h^{\mathrm{out}}} \frac{\boldsymbol{u}_i(m + \varepsilon\psi) - \boldsymbol{u}_i(m)}{\varepsilon}\phi_i^{\mathrm{out}}(x)$$
$$= \lim_{\varepsilon \to 0} \sum_{i=1}^{N_h^{\mathrm{out}}} \frac{\boldsymbol{u}_i(\boldsymbol{m}_1 + \varepsilon\boldsymbol{\psi}_1, \cdots, \boldsymbol{m}_{N_h^{\mathrm{in}}} + \varepsilon\boldsymbol{\psi}_{N_h^{\mathrm{in}}}) - \boldsymbol{u}_i(\boldsymbol{m}_1, \cdots, \boldsymbol{m}_{N_h^{\mathrm{in}}})}{\varepsilon}\phi_i^{\mathrm{out}}(x) \tag{30}$$
$$= \sum_{i=1}^{N_h^{\mathrm{out}}} \left(\frac{\partial \boldsymbol{u}_i}{\partial \boldsymbol{m}_1}\boldsymbol{\psi}_1 + \cdots + \frac{\partial \boldsymbol{u}_i}{\partial \boldsymbol{m}_{N_h^{\mathrm{in}}}}\boldsymbol{\psi}_{N_h^{\mathrm{in}}}\right)\phi_i^{\mathrm{out}}(x)$$
$$= \phi^{\mathrm{out}}(x)(\nabla_{\boldsymbol{m}}\boldsymbol{u})\boldsymbol{\psi},$$

where $\phi^{\text{out}}(x) = (\phi_1^{\text{out}}(x), \ldots, \phi_{N_h^{\text{out}}}^{\text{out}}(x))$ are the finite element basis functions of output function space $V_h^{\text{out}}$.

Then for any test function $v \in V_h^{\text{in}}$, it holds that

$$\langle d^* u(m; p), v \rangle = \langle p, du(m; v) \rangle \quad \text{(by the definition of adjoint operator)}$$
$$= \langle \phi^{\text{out}}(x)(\nabla_{\boldsymbol{m}} \boldsymbol{u}) \boldsymbol{\psi}, \phi^{\text{out}}(x)(\nabla_{\boldsymbol{m}} \boldsymbol{u}) \boldsymbol{v} \rangle \quad \text{(by Equation (30))}$$
$$= \boldsymbol{v}^T (\nabla_{\boldsymbol{m}} \boldsymbol{u})^T M^{\text{out}} (\nabla_{\boldsymbol{m}} \boldsymbol{u}) \boldsymbol{\psi},$$

where $M^{\text{out}}$ is the mass matrix of output function space $V_h^{\text{out}}$, i.e., $M_{ij}^{\text{out}} = \langle \phi_j^{\text{out}}, \phi_i^{\text{out}} \rangle$ for $1 \le i, j \le N_h^{\text{out}}$. Note that if we replace $v$ by the $i$-th finite element basis functions $\phi_i^{\text{in}}$, then $\boldsymbol{v}$ becomes the standard unit vector $e_i \in \mathbb{R}^{N_h^{\text{in}}}$ (with the $k$-th entry one and all others zero). Thus,

$$\langle d^* u(m; p), \phi_i^{\text{in}} \rangle = e_i^T (\nabla_{\boldsymbol{m}} \boldsymbol{u})^T M^{\text{out}} (\nabla_{\boldsymbol{m}} \boldsymbol{u}) \boldsymbol{\psi}, \quad 1 \le i \le N_h^{\text{in}}.$$

Concatenating all the above equations yields

$$\begin{pmatrix} \langle d^* u(m; p), \phi_1^{\text{in}} \rangle \\ \vdots \\ \langle d^* u(m; p), \phi_{N_h^{\text{in}}}^{\text{in}} \rangle \end{pmatrix} = (\nabla_{\boldsymbol{m}} \boldsymbol{u})^T M^{\text{out}} (\nabla_{\boldsymbol{m}} \boldsymbol{u}) \boldsymbol{\psi}.$$

Next, we prove that

$$\begin{pmatrix} \langle \lambda \mathcal{C}^{-1} \psi, \phi_1^{\text{in}} \rangle \\ \vdots \\ \langle \lambda \mathcal{C}^{-1} \psi, \phi_{N_h^{\text{in}}}^{\text{in}} \rangle \end{pmatrix} = \lambda A^{\text{in}} (M^{\text{in}})^{-1} A^{\text{in}} \boldsymbol{\psi}. \tag{31}$$

Indeed, if we let $w = \lambda \mathcal{C}^{-1} \psi$, then similar to the argument in Appendix A.2, we have

$$\lambda A^{\text{in}} (M^{\text{in}})^{-1} A^{\text{in}} \boldsymbol{\psi} = M^{\text{in}} \boldsymbol{w},$$

Note that

$$\langle w, \phi_i^{\text{in}} \rangle = e_i^T M^{\text{in}} \boldsymbol{w}, \quad 1 \le i \le N_h^{\text{in}}.$$

Thus,

$$\begin{pmatrix} \langle w, \phi_1^{\text{in}} \rangle \\ \vdots \\ \langle w, \phi_{N_h^{\text{in}}}^{\text{in}} \rangle \end{pmatrix} = I M^{\text{in}} \boldsymbol{w} = \lambda A^{\text{in}} (M^{\text{in}})^{-1} A^{\text{in}} \boldsymbol{\psi}.$$

Combining (28), (29) and (31) yields

$$\mathbb{E}_{\boldsymbol{m} \sim \nu(\boldsymbol{m})} [(\nabla_{\boldsymbol{m}} \boldsymbol{u})^T M^{\text{out}} (\nabla_{\boldsymbol{m}} \boldsymbol{u}) \boldsymbol{\psi}] = \lambda A^{\text{in}} (M^{\text{in}})^{-1} A^{\text{in}} \boldsymbol{\psi}.$$

# B  Experimental details

## B.1  Governing equations

**Hyperelasticity equation.** We follow the problem setup in [30]. Write $X$ (instead of $x$) for a material point in the domain $\Omega$ and $u = u(X) : \mathbb{R}^2 \to \mathbb{R}^2$ for the displacement of the material point. Under the influence of internal and/or external forces, the material point is mapped to a spatial point $x = x(X) = X + u(X) : \mathbb{R}^2 \to \mathbb{R}^2$. Let $F = \nabla_X x = I + \nabla_X u : \mathbb{R}^2 \to \mathbb{R}^{2 \times 2}$ denote the deformation gradient. For a hyperelasticity material, the internal forces can be derived from a strain energy density

$$W(X, C) = \frac{\mu(X)}{2} (\text{tr}(C) - 3) + \frac{\lambda(X)}{2} (\ln(J))^2 - \mu(X) \ln(J).$$

Here, $C = F^T F$ is the right Cauchy-Green strain tensor, $\text{tr}(C)$ is the trace of matrix $C$, $J$ is the determinant of matrix $F$, and $\mu(X), \lambda(X) : \mathbb{R}^2 \to \mathbb{R}$ are the Lamé parameters which we assume to be related to Young's modulus of elasticity $E(X) : \mathbb{R}^2 \to \mathbb{R}$ and Poisson ratio $\nu \in \mathbb{R}$

$$\mu(X) = \frac{E(X)}{2(1+\nu)}, \quad \lambda(X) = \frac{\nu E(X)}{(1+\nu)(1-2\nu)}.$$

We assume the randomness comes from the Young's modulus $E(X) = e^{m(X)} + 1$. Let $S = 2\frac{\partial W}{\partial C}$ denote the second Piola-Kirchhoff stress tensor. We consider the case where the left boundary of the material is fixed, and the right boundary is subjected to stretching by an external force $t = t(X) : \mathbb{R}^2 \to \mathbb{R}^2$. The strong form of the steady state PDE can be written as

$$\begin{cases} \nabla_X \cdot (FS) = 0, & X \in \Omega, \\ u = 0, & X \in \Gamma_{\text{left}}, \\ FS \cdot n = 0, & X \in \Gamma_{\text{top}} \cup \Gamma_{\text{bottom}}, \\ FS \cdot n = t, & X \in \Gamma_{\text{right}}, \end{cases}$$

where $\Gamma_{\text{left}}, \Gamma_{\text{right}}, \Gamma_{\text{top}}$ and $\Gamma_{\text{bottom}}$ denote the left, right, top, and bottom boundary of the material domain $\Omega$, respectively, and $n$ is the unit outward normal vector on the boundary. Our goal is to learn the operator that maps the parameter $m$ to the displacement $u$. For demonstration, we choose $\bar{m} = 0$, $\delta = 0.4$, $\gamma = 0.04$, Poisson ratio $\nu = 0.4$, and the external force

$$t(X) = \left(0.06\exp(-0.25|X_2 - 0.5|^2), 0.03(1 + 0.1X_2)\right)^T.$$

In practice, we solve the PDE by first formulating the energy $\widetilde{W}$ in the weak form

$$\widetilde{W} = \int_\Omega W \, dX - \int_{\Gamma_{\text{right}}} \langle t, u \rangle \, ds$$

and then solving for $u$ that satisfies the stationary condition, that is, the equation

$$d\widetilde{W}(u; v) = 0,$$

holds for any test function $v$ in the state space. See Figure 3 for the visualization for one parameter-solution pair of the hyperelasticity equation.

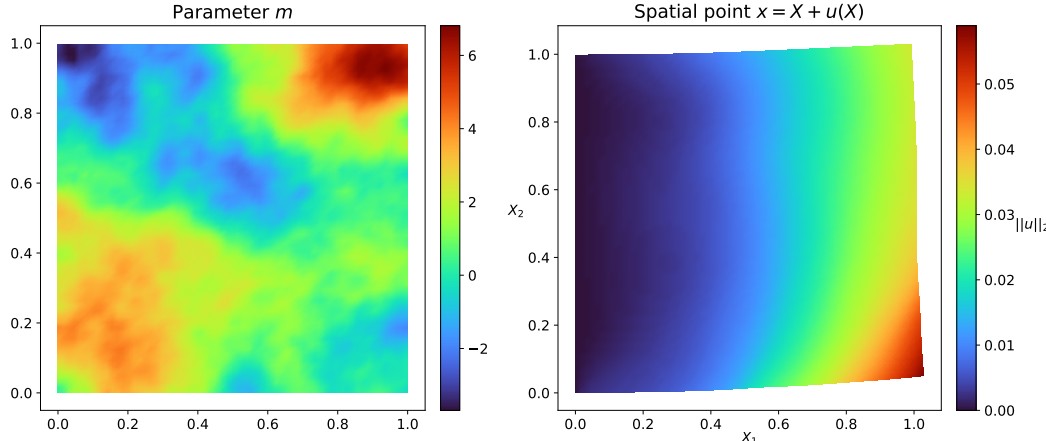

Figure 3: Visualization of one parameter-solution pair of hyperelasticity equation. The color of the output indicates the magnitude of the displacement $u$ (which maps from domain $\Omega$ to $\mathbb{R}^2$) instead of its componentwise function $u_1$ or $u_2$. The skewed square shows locations of any domain point after deformation $X \to x$. See Figures 13 and 14 for $u_1$ and $u_2$.

**Navier–Stokes equations.** Let $u = u(x) \in \mathbb{R}^2$ and $p = p(x) \in \mathbb{R}$ denote the velocity and pressure at point $x \in \Omega = (0,1)^2$. The strong form of the PDE can be written as

$$
\begin{cases}
-\nabla \cdot e^m \nabla u + (u \cdot \nabla)u + \nabla p = 0, & x \in \Omega, \\
\nabla \cdot u = 0, & x \in \Omega, \\
u = (1,0)^T, & x \in \Gamma_{\text{top}}, \\
u = (0,0)^T, & x \in \Gamma \setminus \Gamma_{\text{top}},
\end{cases}
$$

where $\Gamma_{\text{top}}$ and $\Gamma$ denote the left and whole boundary of the cavity domain $\Omega$, respectively. Here, we assume that the randomness arises from the viscosity term $e^m$. Our goal is learn the operator that maps the parameter $m$ to the velocity $u$. For demonstration, we choose $\bar{m} = 6.908$ ($e^{\bar{m}} \approx 10^3$, thus the viscosity term dominates), $\delta = 0.6$, and $\gamma = 0.06$. See Figure 4 for the visualization for one parameter-solution pair of the Navier–Stokes equations.

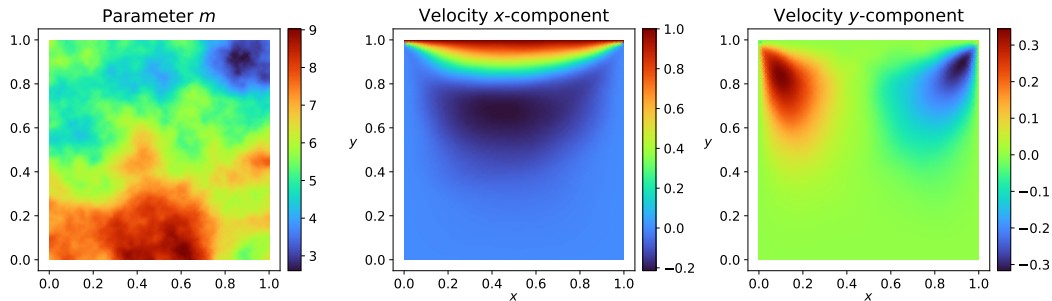

Figure 4: Visualization of one parameter-solution pair of Navier–Stokes equations.

## B.2 Data generation

For all PDEs in this work, we use the class `dolfin.UnitSquareMesh` to create a triangular mesh of the 2D unit square with 64 cells in horizontal direction and 64 cells in vertical direction. For the Darcy flow equation and hyperelasticity equation, we set the direction of the diagonals as 'right', while for the Navier–Stokes equation, we set the direction of the diagonals as 'crossed'. See Figure 5 for a visualization of the unit square mesh with a 10 by 10 resolution.

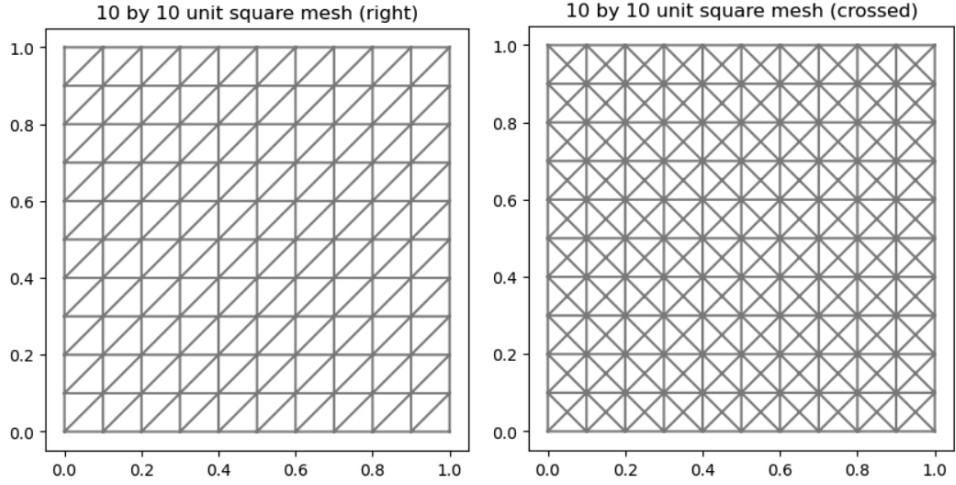

Figure 5: Visualization of the 10 by 10 unit square mesh. Left: diagonal='right'; Right: diagonal='crossed'

We use the class `dolfin.FunctionSpace` or `dolfin.VectorFunctionSpace` to create the finite element function space of input function and output function. For the finite element basis functions, we consider the Continuous Galerkin (CG) family (or the standard Lagrange family) with degree 1 or 2. See Table 6 for details.

Table 6: Configurations of data generation for different datasets

| Dataset | Configurations | | | | | | |
|---------|------|---------------|----------------|----------------------------|----|------------|---|
| | Mesh | $\phi_i^{\text{in}}$ | $\phi_i^{\text{out}}$ | $(N_{\text{train}}, N_{\text{test}})$ | $r$ | $N_{\text{grad}}$ | $s$ |
| Hyperelasticity | $64 \times 64$, right | CG (1D, deg=1) | CG (2D, deg=1) | $(1500, 500)$ | 16 | 16 | 10 |
| Navier–Stokes | $64 \times 64$, crossed | CG (1D, deg=1) | $u$: CG (2D, deg=2), $p$: CG (1D, deg=1) | $(1500, 500)$ | 16 | 16 | 10 |

We generate $N_{\text{train}} = 1500$ and $N_{\text{test}} = 500$ input-output pairs $(m^{(i)}, u^{(i)})$ for training and testing, respectively. We compute first $r = 16$ KLE basis and ASM basis using double pass randomized algorithm with an oversampling parameter $s$ of 10. In the computation of ASM basis, we use $N_{\text{grad}} = 16$ samples for the Monte Carlo estimate of the action of operator $\mathcal{H}$ in Equation (21). In our method, we formulate the different labels into arrays with the shape as follows

- Evaluation labels: $(N_{\text{train}}, N_x, N_u)$
- Derivative $m$ labels: $(N_{\text{train}}, N_x, r, N_u)$

Recall that $N_{\text{train}}$ is the number of functions used for training, $N_x$ is the number of nodes of the mesh, $N_u$ is the dimension of output function, $r$ is the number of reduced basis, and $d$ is the dimension of the domain.

## B.3 Computation of derivative labels and outputs

### B.3.1 Derivative labels $p := du(m; \psi)$ as ground truth

Since the PDE residual $\mathcal{R}(m, u(m)) \equiv 0$ holds for any $m \in V_h^{\text{in}}$, we have

$$\mathcal{R}(m + \varepsilon\psi, u(m + \varepsilon\psi)) = 0, \quad \forall \varepsilon > 0, \psi \in V_h^{\text{in}}.$$

By the Taylor expansion and $\mathcal{R}(m, u(m)) = 0$, we obtain

$$(\partial_m \mathcal{R}(m, u(m)))\varepsilon\psi + (\partial_u \mathcal{R}(m, u(m)))\delta u \approx 0, \tag{32}$$

where $\delta u = u(m + \varepsilon\psi) - u(m)$. Dividing both sides of Eq. (32) by $\varepsilon$ and letting $\varepsilon$ approach 0 yields

$$(\partial_m \mathcal{R})\psi + (\partial_u \mathcal{R})p = 0, \tag{33}$$

where $p := du(m; \psi)$. We solve Eq. (33) for $p$ via its weak form

$$\langle (\partial_m \mathcal{R})\psi, v \rangle + \langle (\partial_u \mathcal{R})p, v \rangle = 0,$$

where $v$ is a test function in $V_h^{\text{out}}$.

**Example.** Consider the (nonlinear) diffusion-reaction equation

$$\begin{cases} -\nabla \cdot (e^m \nabla u) + u^3 = 1, & x \in \Omega, \\ u(x) = 0, & x \in \partial\Omega. \end{cases}$$

Then

- $\mathcal{R} = -\nabla \cdot (e^m \nabla u) + u^3 - 1$
- $(\partial_m \mathcal{R})\psi = -\nabla \cdot (e^m \psi \nabla u)$
- $(\partial_u \mathcal{R})p = -\nabla \cdot (e^m \nabla p) + 3u^2 p$

Thus, $p$ satisfies the linear PDE

$$\langle e^m \psi \nabla u, \nabla v \rangle + \langle e^m \nabla p, \nabla v \rangle + \langle 3u^2 p, v \rangle = 0.$$

Using FEniCS, we can easily derive Gâteaux derivative via automatic symbolic differentiation instead of by hand. In this case, the Python code for representing the weak form of the residual $\langle \mathcal{R}, v \rangle$ and Gâteaux derivatives $\langle (\partial_m \mathcal{R})\psi, v \rangle$ and $\langle (\partial_u \mathcal{R})p, v \rangle$ can be written as

```
import dolfin as dl
```

- `R=(dl.inner(dl.exp(m)*dl.grad(u), dl.grad(v))*dl.dx`
  `+(u**3-dl.Constant(1.0))*v*dl.dx)`

- `dl.derivative(R,m,psi)`

- `dl.derivative(R,u,p)`

### B.3.2 Derivative outputs of neural networks

**DE-DeepONet.** For notation simplicity, we illustrate the case where the input reduced basis is ASM basis and the output function is real-valued. The output of the model is given by

$$\hat{u}(\boldsymbol{m};\theta)(x) = \langle b((\Psi^{\text{in}})^T C^{-1}\boldsymbol{m};\theta_b), t(x;\theta_t)\rangle + \theta_{\text{bias}},$$

where $\Psi^{\text{in}} = (\boldsymbol{\psi}_1^{\text{in}}|\cdots|\boldsymbol{\psi}_{r_{\text{in}}}^{\text{in}}) \in \mathbb{R}^{N_h^{\text{in}} \times r_{\text{in}}}$ are the nodal values of input (ASM) reduced basis functions, $C^{-1}$ is the inverse of the covariance matrix of the Gaussian random field $\nu$ where parameter $m$ is sampled from (Recall that the ASM basis are orthonormal in the inner product with weight matrix $C^{-1}$.) Thus, by the chain rule of derivative, for any test direction $\boldsymbol{\psi}_{\text{test}}$, one has

$$\nabla_{\boldsymbol{m}}\hat{u}(\boldsymbol{m};\theta)(x)\boldsymbol{\psi}_{\text{test}} = \nabla_{\widetilde{m}}\langle b(\widetilde{m}), t(x;\theta_t)\rangle(\Psi^{\text{in}})^T C^{-1}\boldsymbol{\psi}_{\text{test}},$$

where $\widetilde{m} = (\Psi^{\text{in}})^T C^{-1}\boldsymbol{m}$. The Jacobian matrix $\nabla_{\widetilde{m}}\langle b(\widetilde{m}), t(x;\theta_t)\rangle$ can be efficiently computed using, e.g., `torch.func.jacrev`, and further parallelized with `torch.vmap`. If $\boldsymbol{\psi}_{\text{test}}$ is in $\Psi^{\text{in}}$ during model training, we can see that $(\Psi^{\text{in}})^T C^{-1}\boldsymbol{\psi}_{\text{test}}$ becomes a unit vector which frees us the need to compute it, otherwise in the model inference stage when $\boldsymbol{\psi}_{\text{test}}$ is (the nodal values of) a random function sampled from $\nu$, we compute $T = (\Psi^{\text{in}})^T C^{-1}$ and then $T\boldsymbol{\psi}_{\text{test}}$.

**DINO.** The output of the model is given by

$$\hat{u}(\boldsymbol{m};\theta)(x) = \Phi^{\text{out}}(x)\Psi^{\text{out}}f_\theta((\Psi^{\text{in}})^T C^{-1}\boldsymbol{m}),$$

where $\Phi^{\text{out}}(x) = (\phi_1^{\text{out}}(x), \ldots, \phi_{N_h^{\text{out}}}^{\text{out}}(x)) \in \mathbb{R}^{1 \times N_h^{\text{out}}}$ denotes the output finite element basis functions evaluated on point $x \in \Omega$, $\Psi^{\text{out}} = (\boldsymbol{\psi}_1^{\text{out}}|\cdots|\boldsymbol{\psi}_{r_{\text{out}}}^{\text{out}}) \in \mathbb{R}^{N_h^{\text{out}} \times r_{\text{out}}}$ are the nodal values of output (POD) reduced basis functions, similarly for $\Psi^{\text{in}}$ denoting the input (ASM) reduced basis, and $C^{-1}$ is the inverse of the covariance matrix of the Gaussian random field where parameter $m$ is sampled from. Thus, by the chain rule of derivative, for any test direction $\boldsymbol{\psi}_{\text{test}}$, one has

$$\nabla_{\boldsymbol{m}}\hat{u}(\boldsymbol{m};\theta)(x)\boldsymbol{\psi}_{\text{test}} = \Phi^{\text{out}}(x)\Psi^{\text{out}}\nabla_{\widetilde{m}}f_\theta(\widetilde{m})(\Psi^{\text{in}})^T C^{-1}\boldsymbol{\psi}_{\text{test}},$$

where $\widetilde{m} = (\Psi^{\text{in}})^T C^{-1}\boldsymbol{m}$. For fast evaluations given different $x$, $\boldsymbol{m}$, and $\boldsymbol{\psi}_{\text{test}}$, we first compute $T_{\text{left}} = \Phi^{\text{out}}(x)\Psi^{\text{out}}$ for all points $x$ that need to be evaluated and $T_{\text{right}} = (\Psi^{\text{in}})^T C^{-1}$. Next, we compute $J = \nabla_{\widetilde{m}}f_\theta(\widetilde{m})$ using, e.g., `torch.func.jacrev`, and finally compute $T_{\text{left}}JT_{\text{right}}\boldsymbol{\psi}_{\text{test}}$.

**DeepONet & FNO.** Both models receive the full high dimensional parameter $\boldsymbol{m}$, so we compute the directional derivative $\nabla_{\boldsymbol{m}}\hat{u}(\boldsymbol{m};\theta)\boldsymbol{\psi}_{\text{test}}$ using the Jacobian-vector product `torch.jvp` instead of the full Jacobian in DE-DeepONet and DINO. For the coordinates $x$ as additional inputs, we pad zeros to the tensor $\boldsymbol{\psi}_{\text{test}}$ to match the dimension of the input tensor $(\boldsymbol{m}, \{x^{(j)}\}_{j=1}^{N_x})$.

### B.4 Training details

For the DeepONet, we parameterize the branch net using a CNN and the trunk net using a ResNet. For the FNO, we use the package neuraloperator. For the DINO, we use 16 ASM basis functions for the input dimension reduction and 64 POD basis functions for the output dimension reduction, and parameterize the neural network using a ResNet. For the DE-DeepONet, both the branch net and trunk net are parameterized using ResNets. We train each model for 32768 iterations (with the same batch size 8) using an `AdamW` optimizer [34] and a `StepLR` learning rate scheduler (We disable learning rate scheduler for DE-DeepONet). See Tables 7 to 10 for details. When the loss function comprises two terms, we apply the self-adaptive learning rate annealing algorithm from [23], with an update frequency of 100 and a moving average parameter of 0.9, to automatically adjust the loss weights $\{\lambda_i\}_{i=1}^2$ in Equation (5). Additionally, we standardize the inputs and labels before training.

Table 7: Training details for DeepONet

|  | Dataset | |
| --- | --- | --- |
|  | Hyperelasticity | Navier–Stokes |
| branch net | CNN | CNN |
|  | 6 hidden layers | 6 hidden layers |
|  | 256 output dim | 256 output dim |
|  | ReLU | ReLU |
| trunk net | ResNet | ResNet |
|  | 3 hidden layers | 3 hidden layers |
|  | 512 output dim | 512 output dim |
|  | 512 hidden dim | 512 hidden dim |
|  | ReLU | ReLU |
| initialization | Kaiming Uniform | Kaiming Uniform |
| AdamW (lr, weight decay) | $(10^{-3}, 10^{-6})$ | $(10^{-3}, 10^{-5})$ |
| StepLR (gamma, step size) | $(0.6, 10)$ | $(0.7, 10)$ |
| number of Fourier features | 64 | 64 |
| Fourier feature scale $\sigma$ | 0.5 | 1.0 |

Table 8: Training details for FNO & DE-FNO

|  | Dataset | |
| --- | --- | --- |
|  | Hyperelasticity | Navier–Stokes |
| number of modes | (32,32) | (32, 32) |
| in channels | 3 | 3 |
| out channels | 2 | 2 |
| hidden channels | 64 | 64 |
| number of layers | 4 | 4 |
| lifting channel ratio | 2 | 2 |
| projection channel ratio | 2 | 2 |
| activation function | GELU | GELU |
| AdamW (lr, weight decay) | $(5 \times 10^{-3}, 10^{-4})$ | $(5 \times 10^{-3}, 10^{-4})$ |
| StepLR (gamma, step size) | $(0.9, 25)$ | $(0.9, 50)$ |

Table 9: Training details for DINO

|  | Dataset | |
| --- | --- | --- |
|  | Hyperelasticity | Navier–Stokes |
| neural network | ResNet | ResNet |
|  | 3 hidden layers | 3 hidden layers |
|  | 64 output dim | 64 output dim |
|  | 128 hidden dim | 256 hidden dim |
|  | ELU | ELU |
| initialization | Kaiming Normal | Kaiming Normal |
| AdamW (lr, weight decay) | $(5 \times 10^{-3}, 0.0)$ | $(5 \times 10^{-3}, 10^{-12})$ |
| StepLR (gamma, step size) | $(0.5, 50)$ | $(0.5, 50)$ |

Table 10: Training details for DE-DeepONet

| | Dataset | |
|---|---|---|
| | Hyperelasticity | Navier–Stokes |
| branch net | ResNet | ResNet |
| | 3 hidden layers | 3 hidden layers |
| | 128 output dim | 256 output dim |
| | 128 hidden dim | 256 hidden dim |
| | ELU | ELU |
| trunk net | ResNet | ResNet |
| | 3 hidden layers | 3 hidden layers |
| | 256 output dim | 512 output dim |
| | 256 hidden dim | 512 hidden dim |
| | ReLU | ReLU |
| initialization | Kaiming Uniform | Kaiming Uniform |
| AdamW (lr, weight decay) | $(10^{-3}, 10^{-11})$ | $(10^{-3}, 10^{-11})$ |
| disable lr scheduler | True | True |
| number of Fourier features | 64 | 64 |
| Fourier feature scale $\sigma$ | 0.5 | 1.0 |
| $N_x^{\text{batch}}(= \alpha N_x)$ | $422 \approx (0.1 \times 65^2)$ | $422 \approx (0.1 \times 65^2)$ |

Table 11: Number of trainable parameters in each model

| Dataset | # parameters | | | |
|---|---|---|---|---|
| | DeepONet | FNO & DE-FNO | DINO | DE-DeepONet |
| Hyperelasticity | 4.21 M | 17.88 M | 0.04 M | 0.27 M |
| Navier–Stokes | 4.21 M | 17.88 M | 0.15 M | 1.02 M |

## B.5 Convergence plot

Based on the training time (seconds/iteration) of each model in Table 4, we obtain the convergence plots when the number of training samples is either limited ($N_{\text{train}} = 16, 64$) or sufficient ($N_{\text{train}} = 256, 1024$) in Figures 6 to 9 for the hyperelasticity equation, and in Figures 2, 10 and 11 for the Navier–Stokes equations.

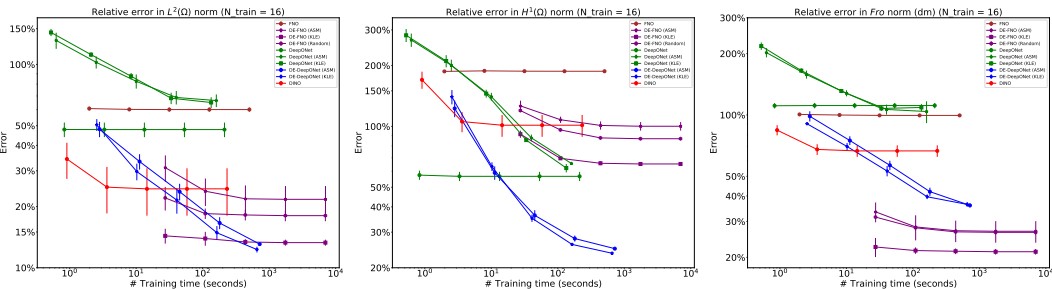

Figure 6: Mean relative errors ($\pm$ standard deviation) over 5 trials versus model training time for the hyperelasticity equations when the number of training samples is 16.

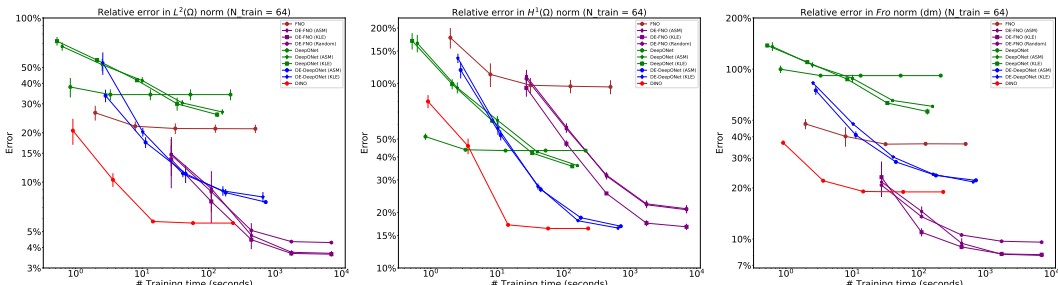

Figure 7: Mean relative errors (± standard deviation) over 5 trials versus model training time for the hyperelasticity equations when the number of training samples is 64.

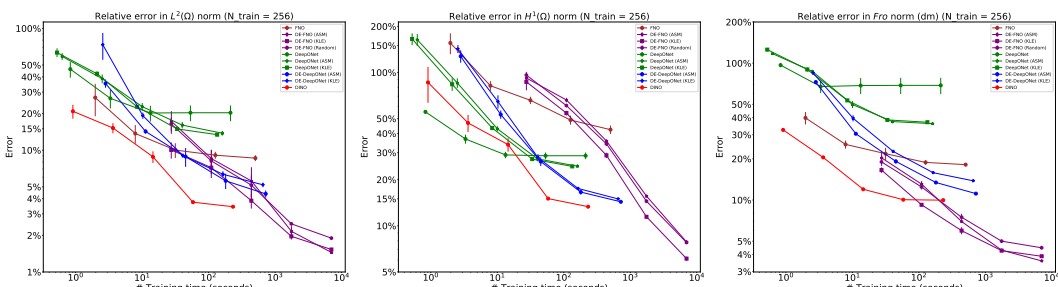

Figure 8: Mean relative errors (± standard deviation) over 5 trials versus model training time for the hyperelasticity equations when the number of training samples is 256.

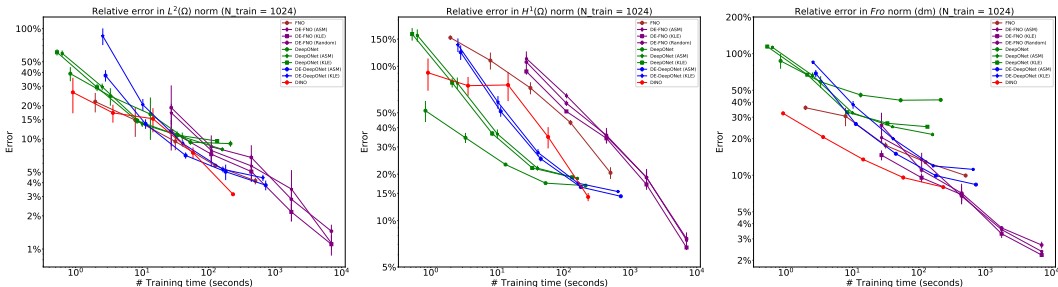

Figure 9: Mean relative errors (± standard deviation) over 5 trials versus model training time for the hyperelasticity equations when the number of training samples is 1024.

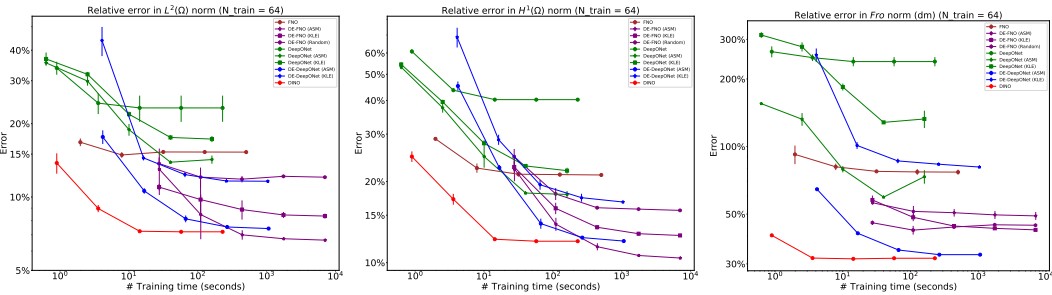

Figure 10: Mean relative errors (± standard deviation) over 5 trials versus model training time for the Navier–Stokes equations when the number of training samples is 64.

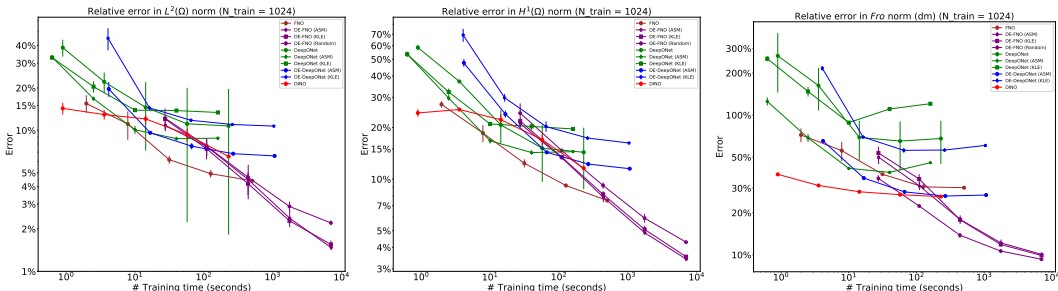

Figure 11: Mean relative errors ($\pm$ standard deviation) over 5 trials versus model training time for the Navier–Stokes equations when the number of training samples is 1024.

## B.6 Output reconstruction error

To measure the error induced by the projection, we define the output reconstruction error as follows

$$\frac{1}{N}\sum_{i=1}^{N}\frac{\|u(P_r m^{(i)}) - u(m^{(i)})\|_{L^2}}{\|u(m^{(i)})\|_{L^2}},$$

where $P_r$ is the rank $r$ linear projector. We provide the plots of the output reconstruction error vs number of reduced basis $r$ using KLE basis and ASM basis in Figure 12. We can see that using the ASM basis results in a lower output reconstruction error than the KLE basis.

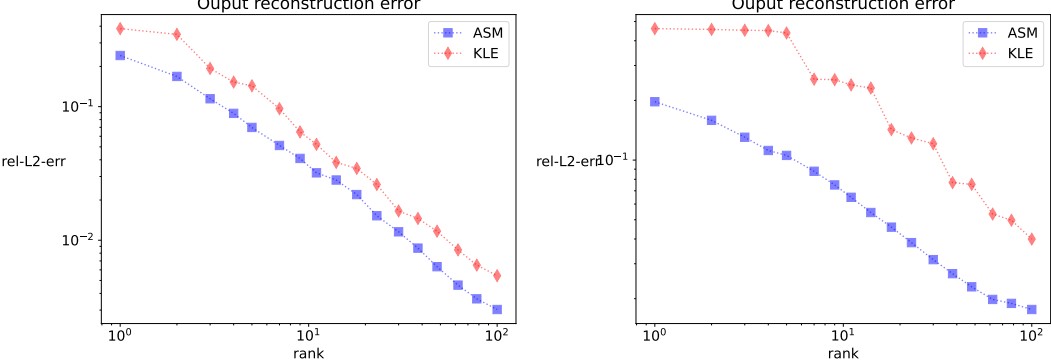

Figure 12: Output reconstruction error using KLE and ASM basis. Left: Hyperelasticity; Right: Navier–Stokes

## B.7 Visualization of the ground truth and prediction

We present the comparisons of the ground truth of solution $u$, model prediction $\hat{u}$ using different methods, and absolute value of their difference $|u(x) - \hat{u}(x)|$ in Figures 13 and 14 for the hyperelasticity equation, and Figures 17 and 18 for the Navier–Stokes equations. In addition, we present the comparisons of the ground truth of the derivative of $u$ with respect to $m$ in the direction of $\omega_1$, denoted as $du(m; \omega_1)$, model prediction $d\hat{u}(m; \psi_1^{\text{in}})$ using different methods, and absolute value of their difference $|du(m; \omega_1) - d\hat{u}(m; \omega_1)|$ in Figures 15 and 16 for the hyperelasticity equation, and Figures 19 and 20 for the Navier–Stokes equations.

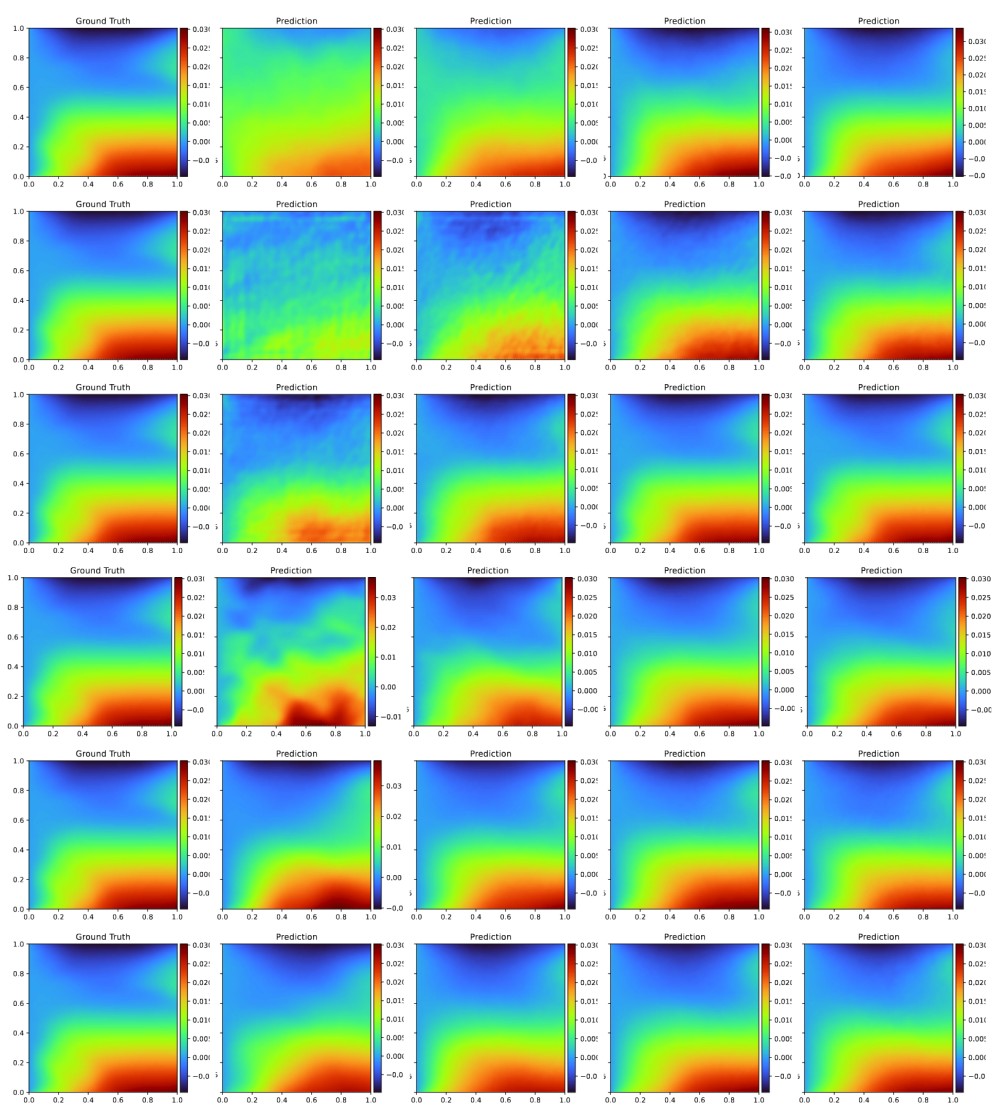

Figure 13: Hyperelasticity. Comparison of the predictions of $u_1$ with (16, 64, 256, 1024) training samples using different methods. Top row –> bottom row: DeepONet, FNO, DE-FNO (Random), DINO, DE-DeepONet (KLE), DE-DeepONet (ASM).

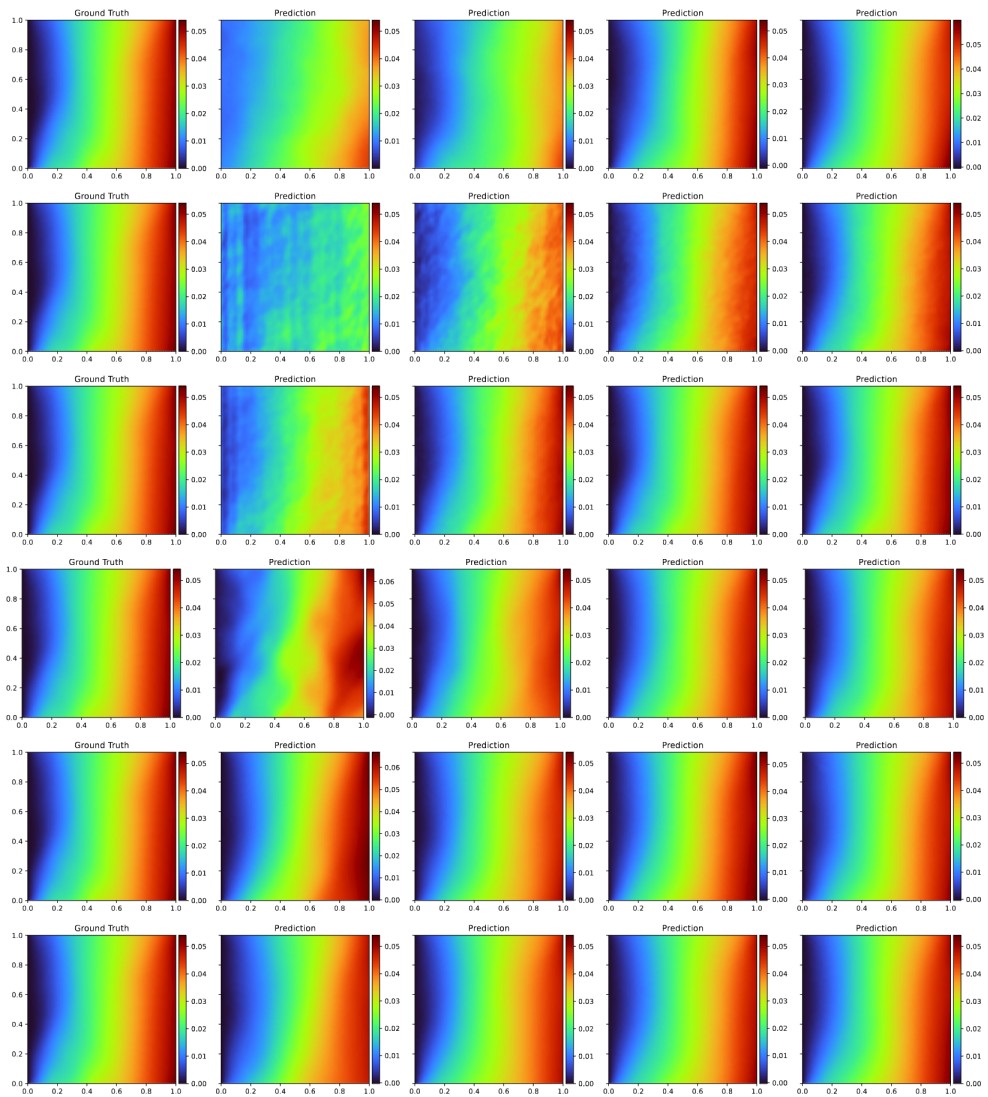

Figure 14: Hyperelasticity. Comparison of the predictions of $u_2$ with (16, 64, 256, 1024) training samples using different methods. Top row –> bottom row: DeepONet, FNO, DE-FNO (Random), DINO, DE-DeepONet (KLE), DE-DeepONet (ASM).

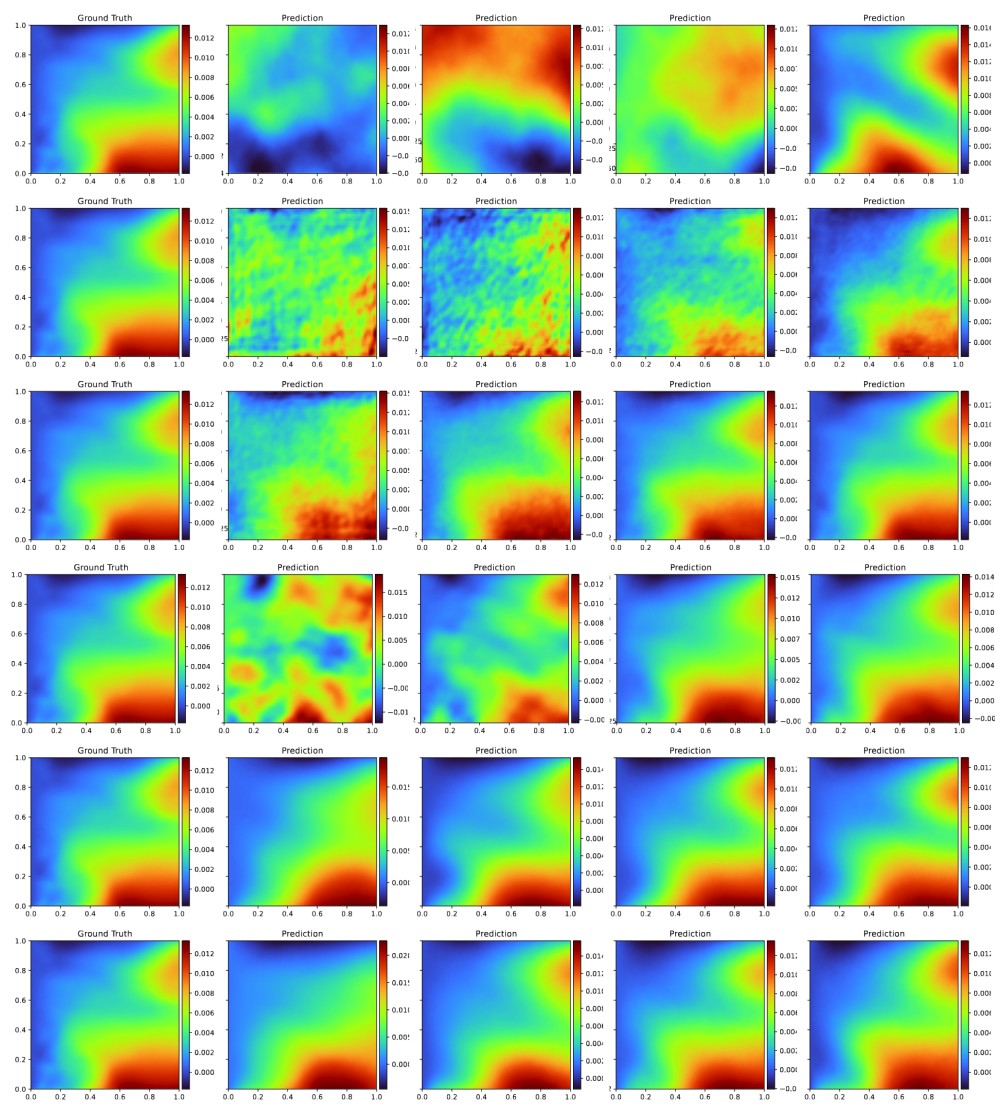

Figure 15: Hyperelasticity. Comparison of the predictions of directional derivative $du_1(m; \omega_1)$ with (16, 64, 256, 1024) training samples using different methods. Top row –> bottom row: DeepONet, FNO, DE-FNO (Random), DINO, DE-DeepONet (KLE), DE-DeepONet (ASM).

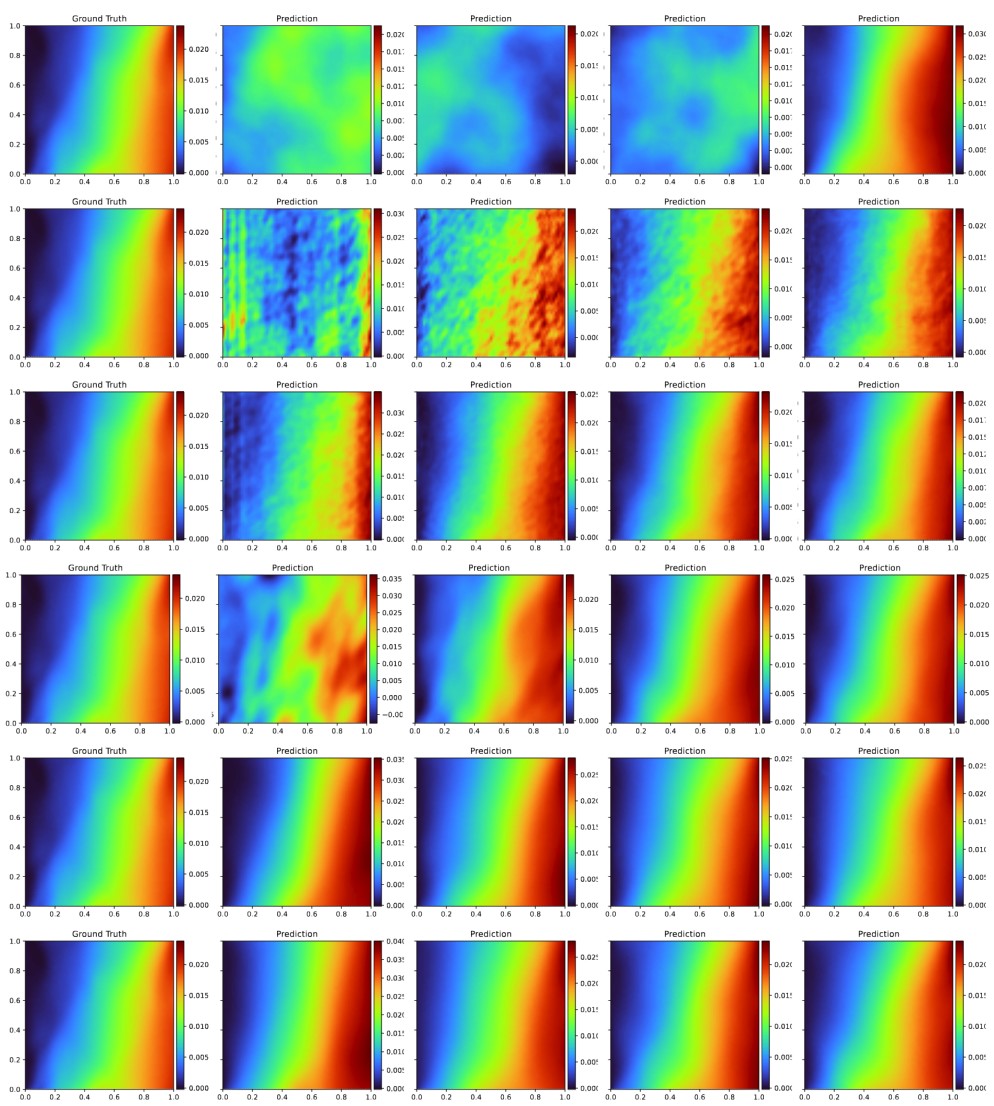

Figure 16: Hyperelasticity. Comparison of the predictions of directional derivative $du_2(m; \omega_1)$ with (16, 64, 256, 1024) training samples using different methods.Top row –> bottom row: DeepONet, FNO, DE-FNO (Random), DINO, DE-DeepONet (KLE), DE-DeepONet (ASM).

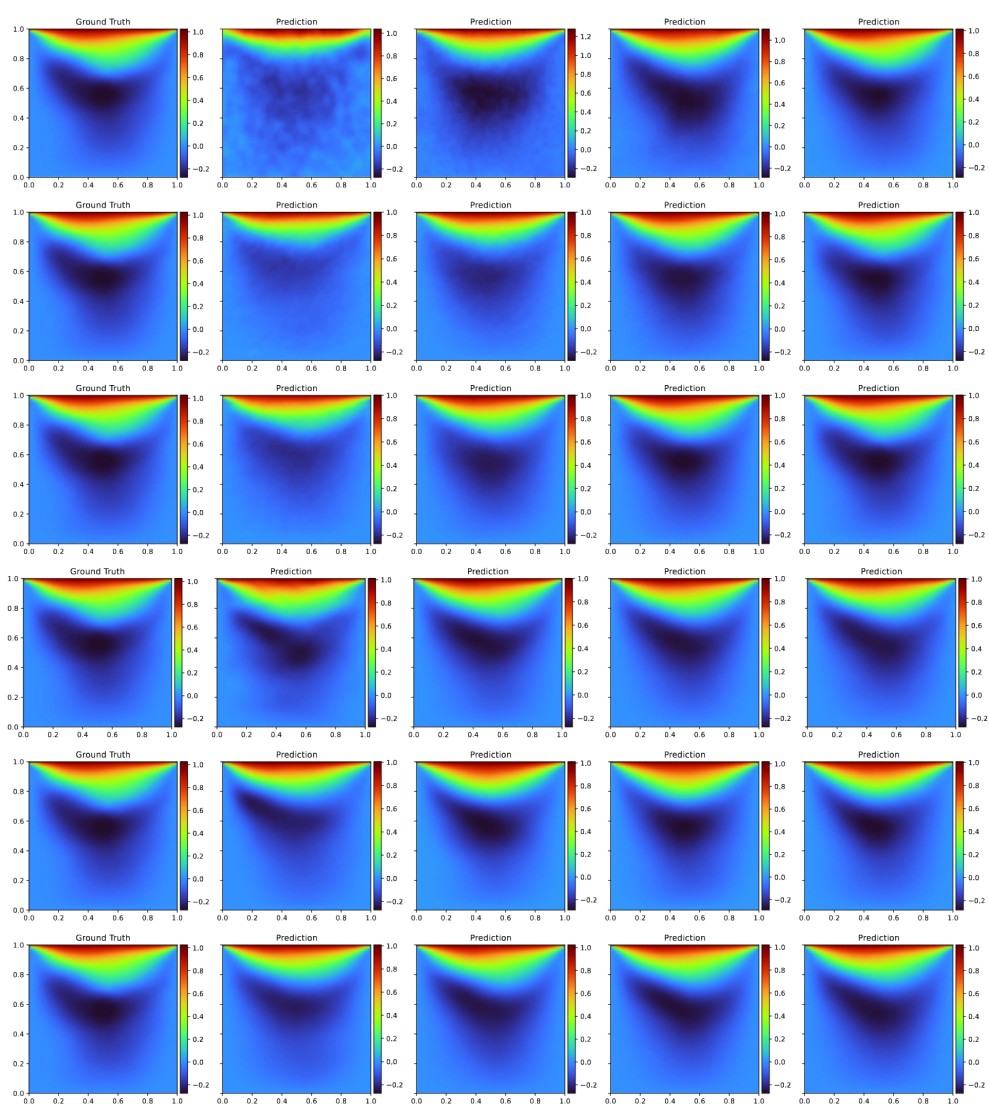

Figure 17: Navier–Stokes. Comparison of the predictions of velocity-x with (16, 64, 256, 1024) training samples using different methods. Top row –> bottom row: DeepONet, FNO, DE-FNO (Random), DINO, DE-DeepONet (KLE), DE-DeepONet (ASM).

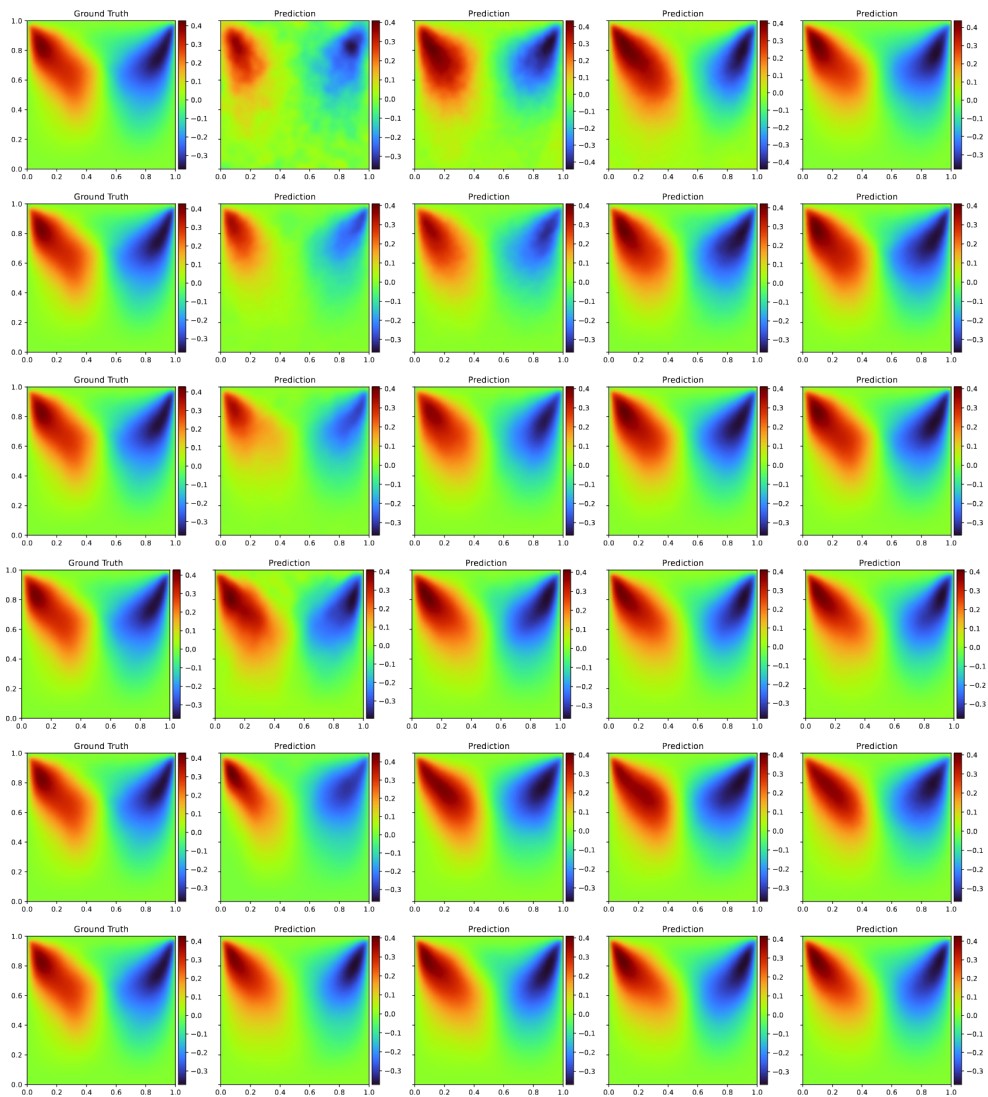

Figure 18: Navier–Stokes. Comparison of the predictions of velocity-y with (16, 64, 256, 1024) training samples using different methods. Top row –> bottom row: DeepONet, FNO, DE-FNO (Random), DINO, DE-DeepONet (KLE), DE-DeepONet (ASM).

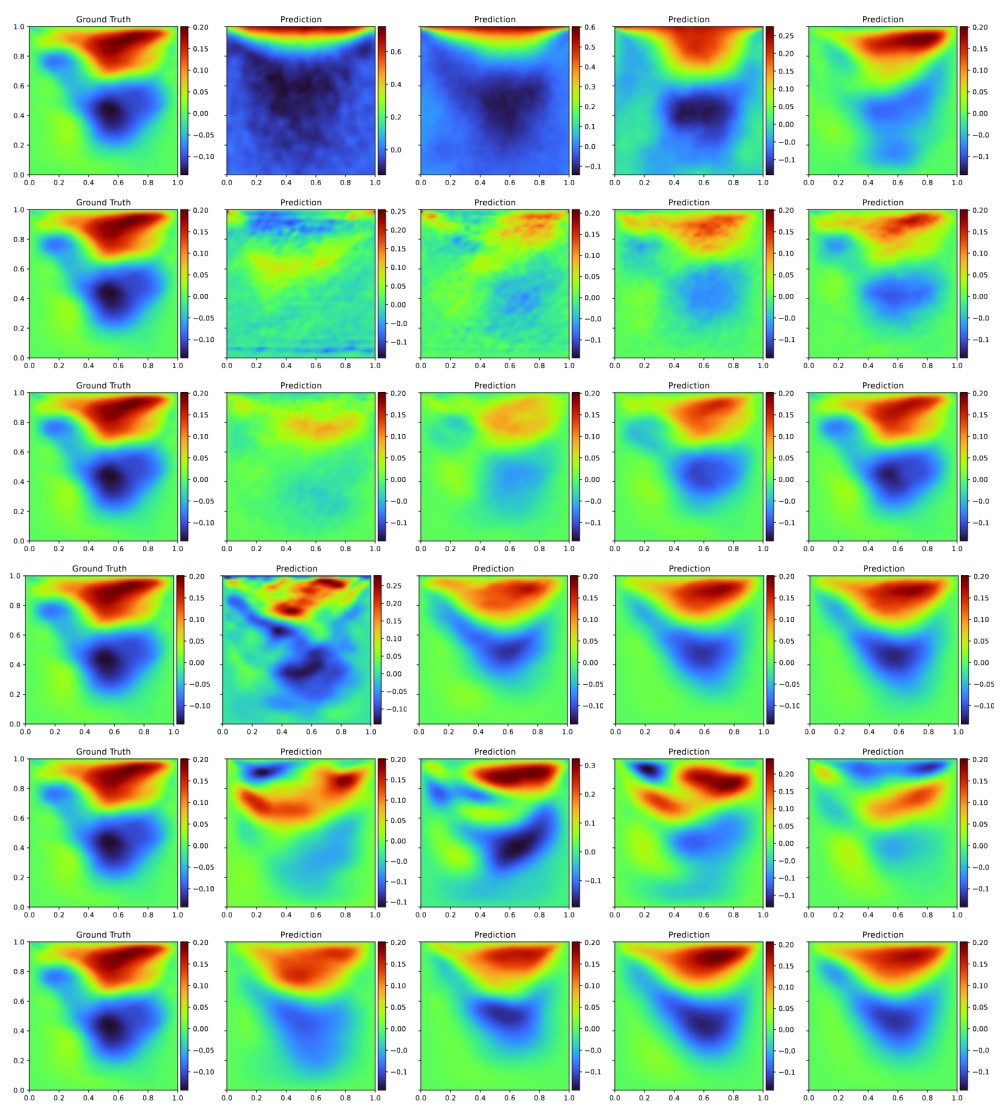

Figure 19: Navier–Stokes. Comparison of the predictions of directional derivative $du_1(m; \omega_1)$ with (16, 64, 256, 1024) training samples using different methods. Top row –> bottom row: DeepONet, FNO, DE-FNO (Random), DINO, DE-DeepONet (KLE), DE-DeepONet (ASM).

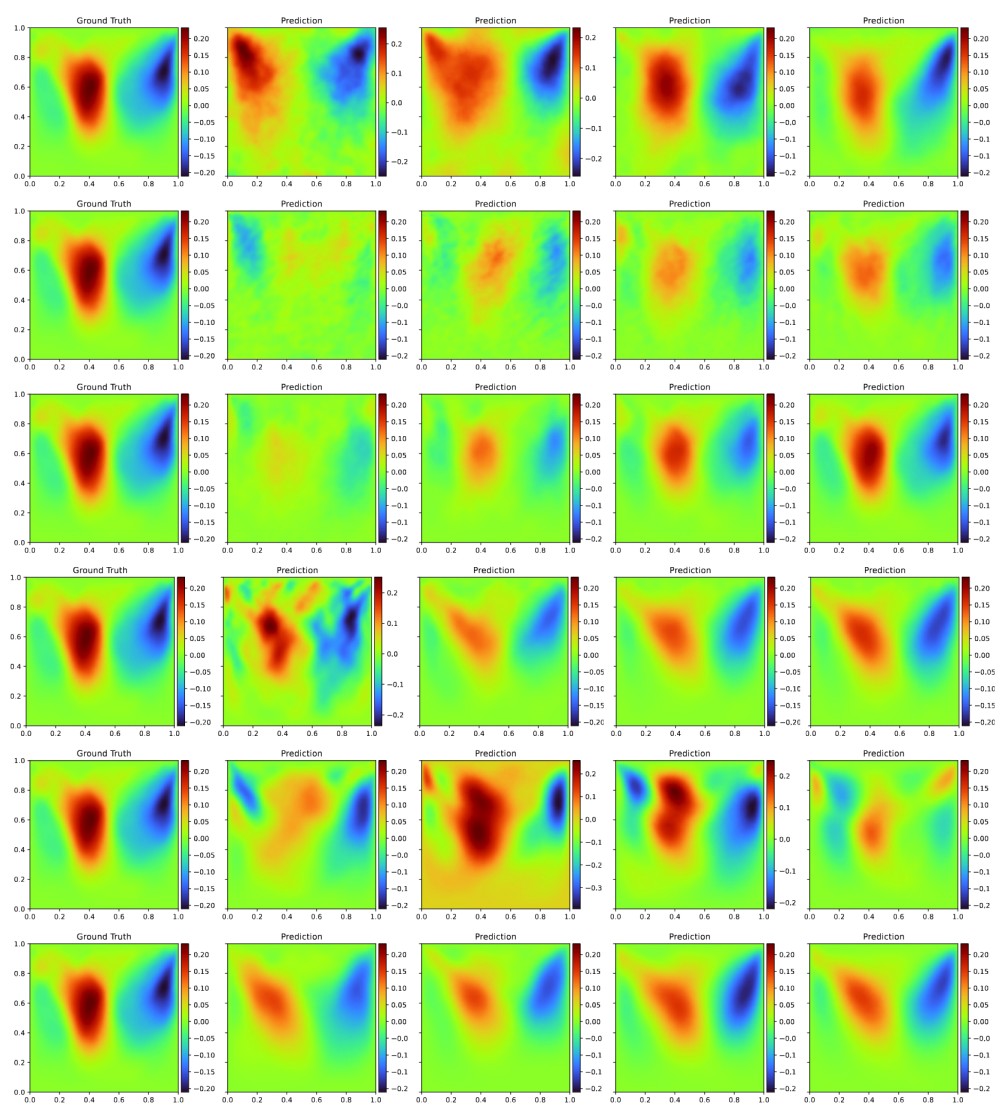

Figure 20: Navier–Stokes. Comparison of the predictions of directional derivative $du_2(m;\omega_1)$ with (16, 64, 256, 1024) training samples using different methods. Top row –> bottom row: DeepONet, FNO, DE-FNO (Random), DINO, DE-DeepONet (KLE), DE-DeepONet (ASM).

