# OpenReview forum: "Derivative-enhanced Deep Operator Network"
_NeurIPS.cc/2024/Conference — NeurIPS 2024 poster_

### Official Review · Reviewer_nyqb · 2024-06-15

**Soundness:** 3
**Presentation:** 4
**Contribution:** 2
**Rating:** 7
**Confidence:** 5

**Summary:**

The authors propose incorporating derivative information into the training of DeepONets to improve predictive accuracy on PDE problems. Instead using an encoder-decoder neural operator architecture as in prior work, the authors use the DeepONet architecture as well as incorporate spatial derivative information on top of functional derivatives previously used.  The authors provide numerical comparisons with other models using both types of derivative training as well as results for two types of input space dimensionality reduction methods.

**Strengths:**

- The paper is very well written, well structured, and with the right level of detail and understanding to describe the setup in the preliminary section without lingering too long. The authors clearly have a deep understanding of all components of this method and convey the steps well in the space given.

- The appendix is thorough, high quality, and provides relevant details to better comprehend and reproduce the results in the main text. I found the visualizations in section B.6 particularly compelling for the low data case with vanilla FNO and DeepONet failing compared to the gradient enhanced models.

- The improvement in dm prediction for control is an important but overlooked metric for neural operators which primary focus on the output solution accuracy despite the gradient information being needed in real-world problems. The authors may want to consider the existing real-world example of DeepONets for optimization in [1], shown in Figure 7 to require gradient information of the objective w.r.t to the input parameters to optimize aerodynamic shapes which could benefit from this methodology which is a strength over accuracy improvement alone.

[1] Shukla, Khemraj, et al. "Deep neural operators as accurate surrogates for shape optimization." Engineering Applications of Artificial Intelligence 129 (2024): 107615.

**Weaknesses:**

-	The cost of generating the dm and dx labels with which to incorporate into training was 2-3 times greater than the cost of solving the PDE itself given Table 1. Of course, as with all neural operators the question is the tradeoff between offline data generation and training versus the benefit of quick online predictions so this cost could be mitigated given the accuracy benefit it provides. Nevertheless, its quite the increase and one may wonder if its worth doing at that point compared to just using the high fidelity FEM solver for N number of problems. Unless N is high enough, the tradeoff looks less appealing here.

-	I would like to see the authors rephrase their findings in the context of the results in B.4 which show that the dx information provides minimal to no benefit compared to the dm training, and it some cases it even makes the accuracy worse than dm alone. This comparison of dx and dm alone should be stated in the main text as it appears to be critical information. Given that beside the solution accuracy, the dm accuracy is vastly more important that the dx accuracy to perform PDE-constrained optimization problems, it begs the question why dx regularization is done at all here. This then begs the question how novel this is compared to DINO which already trained on dm information. The authors claim Sobolev learning is novel on top of the DINO results but it does not appear impactful. The authors need to address this.

-	Following onto the prior point, I am on the edge regarding impact and novelty here. That being said I still think the work is informative and of high enough quality to publish, albeit without as strong of an impact due to the main benefits shown to be in line with the existing method DINO. Looking at DE-DeepONet vs. DINO in Figure 1, there is not much difference, and in Table 2 DINO is shown to be much faster.  I would consider changing my rating if this was adequately addressed.

**Questions:**

-	Could the authors please include total wall-clock time to Figure 2 in addition to per epoch. This would be helpful in comparing to the data cost in Figure 1 which is a total. Additionally, the authors may want to consider including a row for end-to-end data and training time such that the DeepONet and FNO include the PDE solution generation time and DINO and DE-DeepONet include the data for all three loss terms. This way the reader can very easily compare the total cost associated with making predictions with a DeepONet versus DE-DeepONet. I think this would benefit the manuscript since it mitigates the cost of the dm dx label generation.

-	How can the epochs only be 1,000 in B.3? In the original DeepONet paper 50,000 – 500,000 iterations were used. Are all models converged to fairly compare them? What do the convergence plots look like? It would be helpful to see them.

-	Is the vanilla DeepONet and FNO also trained with the dimensionality reduction technique on the input, and if so, which one? Figure 1 shows KLE and ASM for DE-DeepONet, and DINO is stated to use ASM, but what about the baseline models. How is it a fair comparison if the inputs are not the same?

-	The usage of CNN for the trunk and ResNet for the branch should not be hidden in the appendix. The vanilla DeepONet uses a fully-connected NN for both the trunk and branch and that would be assumed here, please mention it in 3.1. Additionally, what justification is there for those choices? The CNN for the trunk makes sense to construct the basis but I’m curious about the ResNet for m?

-	The authors should consider the following papers [2,3]. In [3] the authors train DeepONets the physics-informed (dataless) way using gradient information to obey the governing equation. In [2], the authors train a PINN with gradient-enhanced information which could also be done with the previously mentioned PI-DeepONet in [3]. How might DE-DeepONet methodology be incorporated into these models and in what ways is it distinct from them? It would be nice to see this discussion in the main or appendix text.

[2] Yu, Jeremy, et al. "Gradient-enhanced physics-informed neural networks for forward and inverse PDE problems." Computer Methods in Applied Mechanics and Engineering 393 (2022): 114823.

[3] Wang, Sifan, et al. "Learning the solution operator of parametric partial differential equations with physics-informed DeepONets." Science advances 7.40 (2021): eabi8605.

-	How were the gradients computed using automatic-differentiation (AD)? I do not see the ML package mentioned, a package like Jax has more accurate and substantially faster AD than one like PyTorch, see [4] Table 6 and Figure 15. The authors may want to consider this to improve performance and reduce the overhead cost for dm and dx label generation.

[4] Jagtap, Ameya D., et al. "How important are activation functions in regression and classification? A survey, performance comparison, and future directions." Journal of Machine Learning for Modeling and Computing 4.1 (2023).

**Limitations:**

-	Adequately described in manuscript.

---

> ### Author Rebuttal · Authors · 2024-08-07
>
> Thank you for your recognition of our work and for your constructive feedback. They are valuable to us.
>
> 1. We address the concerns on computational cost and dfference between our method and DINO in the general response. Additionally, we provide the convergence plots in the pdf file.  We hope the updated results can better help evaluate our work.
>
> 2. According to ML community, iterations and epochs are different concepts, even though sometimes they can be equal. One epoch is a complete pass of training data while an iteration is a single update to the model's parameters using a batch of data (typically not the whole training data). Therefore, one epoch equals to ceil(N_{train}/batch size) iterations. In our work, we set batch size 8. When N_train = 1024, epochs = 1000, the model's parameters update (1024/8)*1000=128,000 times, which lies in the range of 50,000 - 500,000 the original DeepONet paper used. To make a more fair comparision, we train all models using different number training samples for the same number of iterations (32768) instead of epochs (1000), and evaluate them on test dataset at milestones: 128, 512, 2048, 8192, 32768 and compute the three test relative errors.
>
> 3. The raw inputs are the same for all models, while in vanilla DeepONet and FNO we feed the network with raw inputs (no dimensionality reduction), and in DE-DeepONet and DINO, we feed the network with post-processed inputs (that is, the reduced inputs by projecting raw inputs into low dimensional linear subspace spanned by KLE or ASM basis).
>
> 4. Thanks for the suggestion. Actually we use CNN for the branch (which receives high dimensional vectors m) and ResNet for the trunk (which receives two dimensional spatical coordiates x). We find that using CNN for the branch yields lower rel-L2-err than using ResNet. Some works also consider this choice (see e.g., Table C.1 in [1]). Since in our problem setup, discretizated parameter m on 2D domain dcan be viewed as a image, it is quite natural to consider using neural network architectures that are originally designed to solve image classification tasks.
>
> [1] Lu, Lu, et al. "A comprehensive and fair comparison of two neural operators (with practical extensions) based on FAIR data"
>
> 5. Thanks for reminding us of the closely related work PI-DeepONet and gPINN. We notice that our dm loss can be directly incorporated into PI-DeepONet's branch net without influencing its trunk net. This could possibly further reduce the requirement of training data. Similar to gPINN, it is possible to enforce the outputs obey not only the residual equal to zero but also the directional derivative of the residual w.r.t. m equal to zero. But this method can be potentially more difficult to implement in practice since the Gateaux derivative of residual is more complex than residual.
>
> 6. We use PyTorch to do AD. We thank the reviewer for mentioning Jax, which we will consider in our later work.

---

> > ### Comment · Reviewer_nyqb · 2024-08-08
> >
> > I thank the authors for their response to my review and particularly the new results provided in the overarching rebuttal. In the context of these items, a number of my points have been addressed (removing dx from training, gaining improvement over DINO with the new FNO + dm results, including convergence plots, etc.) and therefore I am raising my score from 6 $\rightarrow$ 7 under the condition these new (necessary) results make it into the revised manuscript before publication and don't die here.

---

> > > ### Author Response · Authors · 2024-08-08
> > >
> > > We sincerely appreciate your feedback and support. We will definitely incorporate the new results into the revised manuscript.

---

### Official Review · Reviewer_WTPz · 2024-07-11

**Soundness:** 2
**Presentation:** 2
**Contribution:** 3
**Rating:** 6
**Confidence:** 3

**Summary:**

This manuscript introduces a derivative-enhanced deep operator network that utilizes derivative information to improve prediction accuracy.

**Strengths:**

S1) This paper presents a new method for improving the accuracy of approximating the output function for DeepONet, along with its directional derivative relative to the input function and spatial derivatives.

S2) The suggested enhancements improve the performance of the basic DeepONet and perform well in scenarios with limited training samples.

**Weaknesses:**

W1) The baselines appear limited as recent benchmarks like GNOT, LSM, ONO, Transsolver, etc., have ye
t to be compared. Numerous variations of FNO have been introduced, such as FFNO, UNO, CoNO, GFNO, UFNO, etc.

W2) The benchmarks dataset appears to be restricted to datasets for Navier-Stokes equations and hyperelasticity, which are not open-source.

W3) ASM is costly in computation, and derivative information is necessary when the PDE form is unknown. Additionally, it needs to scale better with dimensions.

W4) The proposed method builds upon the DINO paper by introducing additional informed losses.

W5) The new losses were found effective only for DeepOnet and lacked experimental evidence compared to FNO.

**Questions:**

Q1) The primary distinction between DINO and the proposed approach lies in utilizing DeepOnet and spatial derivatives. However, spatial derivatives were also introduced in [1]. Also, have you used the approximation for spatial derivative as used in [1]?

[1] DEEP MULTI-SCALE VIDEO PREDICTION BEYOND MEAN SQUARE ERROR. Michael Mathieu, Camille Couprie & Yann LeCun.

Q2) What is the impact of proposed losses when combined with other operators like FNO? Do they enhance the performance of different operators?

Q3) Could approximating the Frobenius norm along a random direction be analogous to score-based diffusion models?

Q4) Is there a specific reason why adding new loss terms relative to L2 still results in poorer performance than FNO when dealing with hyperelasticity as training samples increase?

Q5) How does incorporating this noisy gradient version aid in training?

**Limitations:**

Yes, the authors adequately addressed the limitations.

---

> ### Author Rebuttal · Authors · 2024-08-07
>
> Thank you for your thoughtful comments. Regarding the weaknesses:
>
> 1. Most of the benchmarks you mention are improved by changing the architecture (or essentially the parameterization method) rather than focusing on adding regularization terms to the loss function, which is the focus of our paper. Also, these benchmarks primarily focus on output solution accuracy. While it is possible to modify the existing code to support dm prediction and dm test error evaluation, it would require a significant amount of effort and time. Thanks for your understanding.
>
> 2. We have not found any public datasets containing dm labels, and these labels cannot be directly computed from input-output pairs; they need to be computed by solving a linear PDE induced by the original PDE. Therefore, we have to compute dm labels by ourselves. We use FEniCS to automate the generation of dm labels, thanks to its supporting for Gateaux derivative. The code will be made available upon publication.
>
> 3. Computing ASM basis is approximately equivalent to solving O(N_{grad}) linear PDEs (regardless the original PDEs is nonlinear or not) together with a generlized eignvalue problem, where N_{grad} is the number of input-output pairs used for the Monte Carlo estimation. As shown in Fig 3 \& 5 in [1], at least in some cases, a small number of samples is sufficient to compute the basis accurately. In our problem setups, we find that setting N_{grad} to 16 already provides a satifactory approximation. Also, we use the double pass randomized algorithm when solving the generalized eigenvalue problem to improve scalability.
>
> [1] Zahm, Oliver, et al. "Gradient-based dimension reduction of multivariate vector-valued functions"
>
> 4. The arguments in the general response provide a more detailed explanation of how DE-DeepONet differs from DINO.
>
> 5. We add experiments to show that adding dm regularization also significantly improve FNO's accuracy. We believe this idea can be applied to other variants that have the same two ends (i.e., input and output) of networks.
>
> Regarding the questions:
>
> 1. For the approximation for spatial derivative, [2] use finite difference to compute both spatial derivative outputs and labels. We use PyTorch's AD to compute outputs and FEniCS's builtin gradient function to compute labels.
>
> [2] Mathieu, Michael, et al. "Deep multi-scale video prediction beyond mean square error"
>
> 2. The results in general response show that dm loss greatly enhance FNO's prediction accuracy.
>
> 3. We observe a certain similarity between our dm loss formulation and the loss function in score-based diffusion models. However, a key difference is that in our work, we use the same neural network to approximate both the solution and its derivatives with respect to input function, whereas in score-based diffusion models, the neural network is trained solely to approximate the derivatives. Also, we estimate the derivative of $u(x_i)$ (where x_i are grid points), which differs from estimate the derivative of log of conditional probability in score-bsaed diffusion models.
>
> 5. We believe that the reason why FNO performs better than DE-DeepONet and DINO when training samples are large enough is mainly due to the use of input dimensionality reduction in DE-DeepONet and DINO (where the linear reduction error cannot be eliminated by increasing traning samples) whereas in FNO we use full inputs.
>
> 6. We do not add noise to any gradients/derivatives involved (including dm, dx, and the gradient of trainable parameters). We are not sure that we fully understand your question. Would you mind explaining your question more clearly? Thank you.

---

> ### Comment · Reviewer_WTPz · 2024-08-09
>
> Thank you for the authors' addressing of my comments, weaknesses, and questions. New experimental results were conducted to help the readers understand the proposed losses more appropriately. Concerning the last question, I wanted to see how proposed losses perform when we have noisy datasets. Does it improve the performance?
>
> Although the paper should be revised with the above information before publication, the author continues to polish the manuscript. Lastly, in the revised version, try to make it a little bit more readable for someone not from a purely mathematical background. So, I am willing to raise the score. Thank you for the responses. I have raised my score and hope to see the new changes incorporated into the revised version.

---

> > ### Author Response · Authors · 2024-08-09
> >
> > We greatly appreciate your positive feedback and support. Considering how dm loss performs with noisy datasets is indeed an interesting and meaningful direction to explore, but we believe it would be more natural to study this within the context of, say, inverse problems, which is beyond the scope of our manuscript (our work only considers enhancing surrogate models for forward operators governed by known PDEs).
> >
> > In terms of readlability, we notice some parts of the manuscript might be too dense or abstract for readers that are not from heavy math backgrounds. When revising the manuscript to includer the new results, we will also pay attention to adding more intuitive explanations (although sometimes at the expense of accuracy), concrete examples, or computational details to help a broader audience understand, particularly in how to derive both DE-DeepONet's (same for FNO+dm) and DINO's dm labels in practice.

---

### Official Review · Reviewer_41ML · 2024-07-12

**Soundness:** 3
**Presentation:** 3
**Contribution:** 2
**Rating:** 5
**Confidence:** 3

**Summary:**

The paper proposes an extension of deep operator networks (DeepONets) enhanced by matching the derivatives of the output function with respect to the input function, for example, the derivatives of the PDE solution w.r.t. the input coefficient. To make the computation tractable, the dimensionality of the input function is reduced via a dimension reduction technique, active subspace method, avoiding expensive Jacobian evaluations.

**Strengths:**

- The paper is written clearly, explaining the relevant preliminaries in a concise way and delivering the core contributions of the proposed method.

- The proposed method leverages well-established classical methods (such as KLE and ASM) for building low-dimensional surrogate and incorporate them into advanced neural operator settings (DeepONet).

- Although not very extensive, the paper present a set of numerical results from experimentation on important benchmark problems.

**Weaknesses:**

- The results of the  empirical evaluation do not seem to be strong enough to make the proposed method look like an effective alternative to the existing method (FNO). The gain obtained at the cost of increased computation seems to be marginal.

- Although KLE or ASM is a standard method for reducing dimensions of a field, the number of required bases depends on smoothness or regularity of the field and, thus, the proposed method could be benefitted in some specific scenarios. Some discussions on where this method could be benefitted and where this method would struggle are required.

- Presenting wall time per epoch is informative, but it would provide a more complete picture if the authors could provide the wall time for training to achieve a certain level of accuracy.

- Regarding the statement “when the training data are limited, the increased computation cost is compensated for a significant reduction of errors”: It would be informative if the authors could provide a summarizing figure presenting a result for varying number of training instances and show that the benefits of the proposed method is more pronounced in the data-scarce regime.

- Although it is not in the main body of the paper, Figure 2 seems to miss the entire information. This omission undermines the completeness of the paper and gives the impression that the paper was prepared hastily.

**Questions:**

- Eq (4). Does the branch net take the reduced input vector as an input? Could the branch net just take the original input and approximate the Frechet derivative using the reduced representation?

- Can the authors describe how the Frechet derivative is computed (the derivative with respect to the reduced input) to collect the ground truth data?

- Is the reason for using the relative error for measuring the loss for preventing numerical issues caused by different numerical scales?

**Limitations:**

- A thorough empirical investigation on in which scenarios this proposed method outperforms baseline (and also the opposite scenarios) would be needed.

---

> ### Author Rebuttal · Authors · 2024-08-07
>
> We appreciate your thoughtful comments. Regarding the weaknesses:
>
> 1. We updated the experimental results as shown in the general response. Our method have much lower test relative errors compared to FNO when the training samples are scarce (N_train = 16 or 64). The main reason our method does not perform well with a large number of traning samples (N_train = 256 or 1024) is due to input dimension reduction, where the reduction error becomes the dominant component of the total error. This can be seen from the fact that relative L2 error (when N_train is large) is quite close to the output construction error at rank = 16 shown in Figure 6 in the appendix. Moreover, we show that the additional dm supervision can also be applied to FNO to enhance its prediction of solution and functional derivatives.
>
> 2. The gains and limitations of our method indeed inherits from the linear reduction techniques such as KLE and ASM. We will include a more detailed discussion about this in the revised version.
>
> 3. We show the plots of test errors vs training time in the pdf file in the general response.
>
> 4. Thanks for your suggestion. We provide one table showing the inference time of each method and one table showing the data generation with different number of training samples.
>
> 5. Figure 2 may be a bit misleading. The color of the output indicates the magnitude to the displacement $u$ (which maps from domain $\Omega$ to $\mathbb{R}^2$), rather than any the component of $u$. And the skewed square shows locations of the any domain point after deformation. This is one of the commonly used visualization methods in studying elasticity problems. To see the componentwise function $u_1$ and $u_2$, please refer to Figure 7 and 8 in the appendix.
>
> Regarding the questions:
>
> 1. In Eq(4), the branch net takes the reduced parameters as inputs. If the branch net takes original input, we believe it is not possible to compute Gateaux derivatives (which we actually use) only using reduced inputs by automatic differentiation (AD). But it is possible to compute the Gateaux derivatives of original inputs using AD, and match the these directives outputs with the same labels used in this work. This approach is similar to using dm labels in FNO.
>
> 2. The high level of computation of Gateaux derivative functions $p=du(m;\psi)$ shown in Eq. (13) can be found in the former part of the proof of Theorem 1. The discretized derivative labels are obtained by evaluating function $p$ on the grid points $x_i$. Note that these labels are a little different from the derivative labels used in DINO. In the revised version, we will add more details about the computation of our labels and DINO's labels.
>
> 3. There are two main reasons we use relative error for measuring the loss. First, we find that it aligns the magnitudes of different loss terms (i.e., evaluation loss and derivative losses), with the loss weights lambda_i also being of the same mangnitude. Intuitively, this allows the network to learn multiple objectives more effectively. Our experiments show that in our problem setups, relative L2 error performs better than MSE. Second, our evaluation metrics use relative error instead of absolute error (which is fairer for comparision of ground truth and prediction), it is natural to train the neural network to optimize according to the same metric (with minimal additional computational cost compared to MSE).

---

> > ### Comment · Reviewer_41ML · 2024-08-12
> >
> > Thank you for providing the clarification and the experimental results. The new information on training/inference time is informative and clears my concern to some extent. I adjusted the score accordingly.

---

> > > ### Author Response · Authors · 2024-08-12
> > >
> > > Thank you for your updated response. We're glad that the new results help address your concern to some extent. When revising the manuscript to include the new results, we will also keep your (and other reviewers') feedback in mind.

---

### Official Review · Reviewer_row2 · 2024-07-22

**Soundness:** 3
**Presentation:** 2
**Contribution:** 2
**Rating:** 6
**Confidence:** 2

**Summary:**

The paper introduces a modified version of DeepONet, termed Derivative-Enhanced DeepONet, by incorporating derivative terms relative to the input function and spatial domain into the loss function. It outlines a practical method for calculating these derivatives, which serve as supplementary supervision terms in the training process. The authors demonstrate enhanced performance over other Neural Operator baselines with this approach, particularly in low-data scenarios, using two datasets centered on hyperelasticity and Navier-Stokes equations.

**Strengths:**

- The paper is engaging and well-motivated.
- The methodology presented appears novel and is supported by solid theoretical underpinnings. It demonstrates consistent enhancements across various metrics, particularly with the hyperelasticity dataset.

**Weaknesses:**

- As a reviewer with limited expertise in functional analysis, I found the sections detailing the supplementary supervision terms challenging to comprehend. A more introductory overview explaining how the derivative ground truths are derived would be beneficial for clarity and accessibility.

**Questions:**

- Could you provide a higher-level explanation of how the derivative labels are calculated? Are they derived from the underlying equation?
- On line 325, you mention reducing training costs by introducing additional derivative losses in the later stages of training. Do you have any data or figures to support this claim?
- Why does supervision of the derivatives become less significant as more data becomes available?

**Limitations:**

Limiatations have been addressed

---

> ### Author Rebuttal · Authors · 2024-08-07
>
> Thank you for your recognition of our work. We'd like to address the questions:
>
> 1. The compution of derivative labels $p=du(m;\psi)$ is equivalent to solving Eq.(13). The reviewer can find more details in the former part of the proof Theorem 1. The discretized derivative labels $p(x_i)$ are obtained by evaluating p on the grid points $x_i$. Note that these labels are a little different from the derivative labels used in DINO. In the revised version, we will add more details about the computation of our labels and DINO's labels.
>
> 2. Although we currently do not have extensive experiments in this direction of improvement, we provide a small test. We train DE-DeepONet using dm loss only at the later 10% epoch of total epochs (denoted as DE-DeepONet later) on Navier--Stokes equations. The following are comparisions of the test relative error when using 16 samples to train DE-DeepONet/DeepONet for 32768 iterations.
>
> method            | rel-L2-err   | rel-H1-err | rel-Fro-err |
> ------------------|   -----------| -----------| ------------|
> DE-DeepONet (ASM) |   9.10 %     |   13.01 %   |     39.76 %  |
> DE-DeepONet later (ASM) |   9.49 % |        13.45 %     |     40.87 %  |
> DE-DeepONet (KLE) |   13.75 %        |     18.03 %  |   78.70 %    |
> DE-DeepONet later (KLE) |   19.11 % |       25.15 %      |       136.30 %   |
> DeepONet                 |  27.88 %    |    26.70 %         |     130.79 %   |
>
> We can see that DE-DeepONet later (ASM) has almost the same accuracy as DE-DeepONet (ASM). However, when replacing the ASM basis with the KLE basis, the significant increase in relative error makes this technique less appealing -- it may be necessary to train with dm labels for more epochs.
>
> 3. Intuitively, derivatives can be viewed as sensitivity information for the neighborhood of each input-output pair. In other words, if we slightly perturb the input, we can predict how the output will change accordingly. It's akin to enhancing the training dataset from (input point, output point) to (input neighborhood, output neighborhood), where each point represents a function. When more (input point, output point) pairs are available, the neighborhood information can be revealed by the newly added nearby points, making the derivatives less useful.

---

> > ### Comment · Reviewer_row2 · 2024-08-13
> >
> > I thamk the authors for their clarifications and additional experiments, which help me understand the paper better and answer my questions. I have raised my score from 5 to 6.

---

> > > ### Author Response · Authors · 2024-08-13
> > >
> > > We thank the reviewer for the updated response!

---

### Author Rebuttal · Authors · 2024-08-07

We are sincerely grateful to the reviewers for dedicating their time and effort to reviewing our work and providing helpful feedback.

Compared to DINO, although the DeepONet architecture (and its formulation of dm loss) requires longer training time, it offers the following advantages

- Much shorter inference time of $du(m;\psi)(x)$. The additional trunk net (which receives spatial coordinates) allows us to quickly query the sensitivity of output function $u$ at any point $x$ when input function $m$ is perturbed in any direction $\psi$. While DINO can only provide the derivative of the output coefficients respect to the input coefficients (we call reduced dm), in order to compute the sensitivity at a batch of points, we need to post process the reduced dm by querying the finite element basis on these points and computing large matrix multiplications. We provide more details in the revised version.

- Greater flexibility and potential for improvements. Although both DeepONet and DINO approximate solution by a linear combination of a small set of functions, these functions together in DeepONet is essentially the trunk net, which is "optimized" via model training, whereas in DINO, they are POD or derivative-informed basis precomputed on training samples. When using DINO, if we encounter a case where the training samples not enough to accurately compute the output basis, the large approximation error between the linear subspace and solution manifold will greatly restrict the model prediction accuracy, no matter how we modify the underlying ResNet/MLP and/or loss function. And the reduced dm labels only supports linear reduction of output. However, it is possible that we can further improve DeepONet by, e.g., adding physical losses (to enhance generalization performance) and Fourier feature embeddings (to learn high-frequency components more effectively) on the trunk net [1] and replacing the  inner product of the outputs of two networks by more flexible operations [2] [3] (to enhance expressive power). The dm loss formulation of our work differs nontrivally by DINO's, but it is broadly suitable any network architecture that has multiple subnetworks, where at least one of them receives high-dimensional inputs.

[1] Wang, Sifan, et al. "Learning the solution operator of parametric partial differential equations with physics-informed DeepONets."

[2] Pan, Shaowu, et al. "Neural Implicit Flow: a mesh-agnostic dimensionality reduction paradigm of spatio-temporal data"

[3] Hao, Zhongkai, et al.  "GNOT: A General Neural Operator Transformer for Operator Learning"

We made an update to the experimental results. The modifications include

1. Adding three models -- FNO trained with dm loss along ASM, KLE and Random (sampled from same Gaussian random field of parameter m) directions -- into comparision.
2. Removing dx loss in training. Our experiments show that dx labels typically do not help improve model's prediction accuracy if dm labels are used in training.
3. Fixing iterations (32,768) instead of epochs (1000) for model training, no matter which model or how many training samples are used. We compute the test relative errors at iteration milestones: 128, 512, 2048, 8192, 32768.
4. Adjusting the hyperparameters of DE-DeepONet and DINO. We double their widths and halving their depths. We disable the learning rate scheduler step_LR. For the DE-DeepONet, we disable dx loss and add Fourier features embedding (Gaussian RFF mapping) [4] into trunk net.

[4] Tancik, Matthew, et al. "Fourier features let networks learn high frequency functions in low dimensional domains"

In the file **rebuttal_rel_err.pdf**, we provide eight new plots of relative test errors on 500 test samples on Navier--Stokes equations in L2 norm and Fro norm (H1 norm is similar to L2 norm, we omit it here due to space limitation) versus different number of iterations and total training time when using 16 or 256 training samples (representative of limited data and sufficient data scenarios). We can see that when N_train=16, DE-DeepONet (ASM) and DINO have much lower error compared to FNO, FNO+dm and DeepONet. However, given 256 training samples and sufficient training time, FNO+dm achieves the lowest test error. The choice of perturbation directions (ASM, KLE, and Random) does not make a significant difference. Furthermore, FNO+dm consistently outperforms the vanilla FNO, although it requires much longer training time.

We also provide the total wall clock time of each model inferencing on **500** test samples of **both solution and dm in 128 random directions** on a single GPU and a single CPU (which is needed for post processing of outputs in DE-DeepONet and DINO). Note that the time for predicting dm dominates the total inference time.

| Model | Inference time (seconds) |
|-|-|
|FNO|53|
|DeepONet|4|
|DE-DeepONet|47|
|DINO|415|
|Numerical solver (16 CPUs)|1103|

The reason why DINO has much more inference time compared to DE-DeepONet is due to the post processing the reduced dm requires very large matrices multiplications (one of the dimensions is the degree of freedom of high fedility solution, which in our case is 66050). And this is even an overoptimistic estimate since we exclude the time for computing a repeated used large matrix, i.e., the output finite element basis functions evaluating on gird points and its matrix multiplication with the nodal values of output reduced basis.

Finally, we show the total wall clock time of data generation of our DE-DeepONet (we only includes the major parts -- computing high fidelity solution, ASM basis and dm labels [16 directions]) when N_train = 16, 64, 256, 1024 using **16** CPU processors.

| N_train | Data generation (s) |
|-|-|
|16|17|
|64| 38|
|256|125|
|1024|470|

We hope the above figures and tables provide readers with a more clear view of the offline and online cost vs accuracy.

---

### Comment · Area_Chair_W6BS · 2024-08-08

Dear reviewers,

could you have a look at the authors response and comment on them if you have done so, yet.

thanks in advance

your area chair

---

### Decision · Program_Chairs · 2024-09-25

**Decision:**

Accept (poster)

**Comment:**

The paper introduces a modified version of DeepONet, termed Derivative-Enhanced DeepONet, by incorporating derivative terms relative to the spatial domain into the loss function. The paper use KL expnasion/active subspace methods to approximate the derivative so that all the computation can be implemented in the function space. This paper implements solid experiments and showed the perfomance boost of the proposed loss. The newly introduced trunknet alows faster inference net. This is definitely a solid paper. Thus I vote for acceptness.